# RETHINKING JEPA: COMPUTE-EFFICIENT VIDEO SSL WITH FROZEN TEACHERS

**Xianhang Li**[*]
Apple

**Chen Huang**
Apple

**Chun-Liang Li**[†]
Apple

**Eran Malach**
Apple

**Josh Susskind**
Apple

**Vimal Thilak** [‡]
Apple

**Etai Littwin**[‡]
Apple

## ABSTRACT

Video Joint Embedding Predictive Architectures (V-JEPA) learn generalizable off-the-shelf video representation by predicting masked regions in latent space with an exponential moving average (EMA)-updated teacher. While EMA prevents representation collapse, it complicates scalable model selection and couples teacher and student architectures. We revisit masked-latent prediction and show that a frozen teacher suffices. Concretely, we (i) train a target encoder with a simple pixel-reconstruction objective under V-JEPA masking, then (ii) freeze it and train a student to predict the teacher's latents on masked regions. This leads to a two-stage, unregularized scheme that we refer to as SALT (Static-teacher Asymmetric Latent Training). SALT decouples optimization into pixel reconstruction (teacher) and masked latent prediction (student), increasing transparency, efficiency, and scalability while preserving the ability of representation to generalize under frozen evaluation. Empirically, our student models outperform recently proposed V-JEPA 2 encoders under frozen backbone evaluation across diverse benchmarks. They are also more compute-optimal: at matched pretraining FLOPs, our method achieves higher probing accuracy, and its scaling curves dominate V-JEPA's accuracy–FLOPs Pareto frontier. Finally, we find that student quality is remarkably robust to teacher quality: high-performing students emerge even with small, sub-optimal teachers. This points to a compute budget allocation that should overwhelmingly favor the student. These results position SALT as a simple, scalable, and compute-efficient alternative to EMA-based self-distillation for video representation learning.

## 1 INTRODUCTION

Self-supervised learning (SSL)-based methods have emerged as a standard approach for representation learning in computer vision. These methods pretrain neural networks that use vast amounts of image (He et al., 2021; Assran et al., 2023; Caron et al., 2021; Oquab et al., 2024; El-Nouby et al., 2024) or video (Tong et al., 2022; Wang et al., 2023b; Bardes et al., 2024; Assran et al., 2025) data to learn backbones that have been shown to work well on many downstream tasks. Among these methods, Joint Embedding Predictive Architecture (JEPA)-based methods (LeCun, 2022) have demonstrated a strong ability to learn powerful semantic features that perform well on downstream image (I-JEPA) (Assran et al., 2023) and video (V-JEPA) (Bardes et al., 2024; Assran et al., 2025) tasks.

As concrete instantiations of the Joint Embedding Predictive Architecture (JEPA), I-JEPA (Assran et al., 2023) and V-JEPA (Bardes et al., 2024; Assran et al., 2025) are masking-based pretraining methods that learn powerful semantic representation by predicting masked-out portions of the input in a learned embedding space. Specifically, these methods consist of a context (student)

---

[*]Work done during internship at Apple.

[†]Work done while at Apple.

[‡]Equal last author contribution.

encoder and a predictor that are trained to make predictions that match the embeddings provided by a target (teacher) encoder. While powerful, the JEPA family of models are often complex, hyperparameter-brittle, and use an uninformative loss metric that is a poor proxy for representation quality, requiring practitioners to rely on other more downstream-predictive metrics (Agrawal et al., 2022; Garrido et al., 2023; Thilak et al., 2024). These issues stem from the core JEPA design: because student and teacher representation co-evolve, trivial collapsed solutions with near-zero loss exist, and must be avoided. To prevent representation collapse, these models are implicitly regularized using the self-distillation approach pioneered by BYOL (Grill et al., 2020). Namely, the stop-gradient operation is applied on the target encoder, and its weights are updated by an exponential moving average (EMA) copy of the student weights, according to some EMA scheduler. It is worth mentioning that other variants of joint-embedding SSL models utilize more explicit regularizers to prevent collapse (Zbontar et al., 2021; Bardes et al., 2021).

In this paper, we present a comprehensive empirical study that challenges the common assumption that involved collapse prevention mechanisms are required for learning high-quality semantic features. Specifically, we show that a *dynamic* teacher is unnecessary, and that stable, high-quality targets needed to optimize the student model can be obtained in a more efficient manner with a frozen encoder. This design obviates the need for both the EMA update and the stop-gradient, streamlining the self-distillation process and reducing implementation complexity. We start with a simple two-stage pretraining scheme: (i) train a target encoder with a pixel-reconstruction objective under V-JEPA–style masking; (ii) freeze this encoder and train a student with the JEPA objective to predict the teacher's latents on masked regions (Bardes et al., 2024). Prior work has explored using pretrained frozen encoders as masked-prediction targets (Wang et al., 2023c; Li et al., 2023), but typically assumes access to strong teachers and often relies on fine-tuning the student to realize the benefits. In contrast, our study provides a fair and direct comparison against strong baselines, including V-JEPA 2, on larger datasets and models; we dub our method SALT (Static-teacher Asymmetric Latent Training) and show that:

1. **Small, "sub-optimal" teachers suffice.** High-quality semantic features competitive with state-of-the-art under frozen evaluation protocols can be learned from much smaller and cheaper teachers. Using the strongest available pretrained encoders is unnecessary and yields at most marginal gains for the student.

2. **Compute efficiency.** Our two-stage design is more compute-efficient than EMA-based self-distillation (e.g., V-JEPA): at matched **FL**oating **P**oint **O**perations (FLOPs) and wall-clock, and even when accounting for the cost of training the teacher, our method achieves a better accuracy–FLOPs trade-off[1].

3. **Interpretable model selection.** Our design yields a student loss that provides an informative, training-time metric that correlates strongly with downstream accuracy under the frozen-backbone protocol, in contrast to EMA-based methods that require proxy heuristics for model selection.

Taken together, our results suggest that elaborate online student-teacher dynamics and EMA-based collapse prevention machinery may be unnecessary for learning high-quality representation.

## 2    METHOD OVERVIEW

We first review video-based JEPA models that include both V-JEPA and V-JEPA 2, and then describe our simple approach named SALT for representation learning from videos. Note that V-JEPA 2 uses the same pretraining method described in V-JEPA but employs updated hyperparameters so our method review applies to both models.

### 2.1    V-JEPA

V-JEPA employs a masked prediction objective: the context encoder–predictor reconstructs masked regions from visible frames in a learned representation space, while an EMA-updated target encoder

---

[1]FLOPs and total number of training steps are used interchangeably to refer to compute. Appendix G includes an explanation for this choice

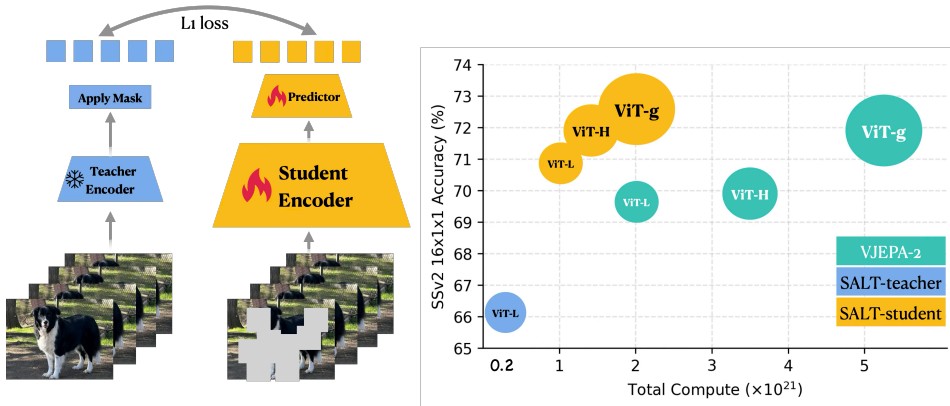

Figure 1: **(Left) SALT Stage 2:** Frozen-teacher, learnable student and predictor. The frozen teacher encoder is obtained via **Stage 1** (not pictured above) by training using a pixel reconstruction objective. The student and predictor are jointly optimized to learn representation from video in Stage 2 using a latent space prediction objective. **(Right):** SALT's compute-accuracy curve dominates V-JEPA 2.

supplies the supervision. Following the notation used by Bardes et al. (2024), the latent space prediction objective can be written:

$$\min_{\theta,\phi} \mathbb{E}_{x,y} \|g_\phi(f_\theta(x), \delta y) - \textbf{stop\_grad}(\bar{f}_\theta(y))\|_1 \tag{1}$$

where $x$ and $y$ denote two disjoint regions of the input, $f$, $\bar{f}$ and $g$ denote the encoder, target encoder and predictor respectively, **stop_grad** denotes the stop-gradient operation and $\delta y$ denotes the spatio-temporal positions of missing regions in the input that act as context for the predictor.

## 2.2 STATIC-TEACHER ASYMMETRIC LATENT TRAINING (SALT) - A SIMPLIFIED VIDEO REPRESENTATION LEARNING METHOD

The V-JEPA method uses a self-distillation approach incorporating stop-gradient and exponential moving average (EMA). This approach requires properly setting the associated hyperparameters to prevent representation collapse. In this work, we advocate for an alternative solution, which we refer to as SALT, that does not require the use of EMA operation. Specifically, we simplify the architecture by breaking down video representation learning into two steps:

- **Stage 1 -** The teacher or target encoder is trained to optimize a pixel reconstruction objective in Stage 1. The objective is identical to the objective used in VideoMAE (Tong et al., 2022). However, our Stage 1 method differs from VideoMAE as we use a more efficient masking scheme, the details of which are described in Section 5.2.

- **Stage 2 -** The weights of the teacher from Stage 1 are frozen and used to train a student and predictor network as shown in Figure 1. The JEPA objective, described by Equation (1), is used to optimize the student and the predictor.

The simplification described above results in two loss objectives that are proper loss functions which are easier to interpret in practice, and are immune to representation collapse by design. This stands in contrast to V-JEPA's objective in Equation (1), which is difficult to interpret due to its self-distillation nature that, in turn, necessitates the use of surrogate metrics (Garrido et al., 2023; Thilak et al., 2024). Moreover, our two-stage approach completely decouples the teacher and student architectures, unlocking considerable compute efficiency gains by utilizing small teachers to train larger students, as shown in Figure 1 and Table 1. We observe from Figure 1 that SALT shows a remarkable improvement over V-JEPA 2 on Something-Something-v2 (SSv2) (Goyal et al., 2017), which is a temporal understanding task. Furthermore, Table 1 shows that a smaller but noticeable improvement is observed on Kinetics-400 (Kay et al., 2017), an appearance understanding benchmark. We describe the experimental setup in Section 3 and discuss results in detail in Section 4.

### 2.2.1 SALT DESIGN PRINCIPLES

SALT follows the contemporary trend toward simple, principled architectures and objectives, avoiding elaborate engineering. We provide a simple recipe to train the teacher in **Stage 1** with method that we call V-Pixel that uses a pixel reconstruction objective along with the multi-block masking method described in V-JEPA [2]. The decoupled design of SALT allows us to study the role of architecture and dataset choices for training the teacher and student in a granular manner. We uncover a surprising finding that a high performing teacher, as measured by its downstream performance, is not necessary to train a high-quality student. As we show in Section 5.1, Section 5.3 and Section 5.4, the student's ultimate quality is surprisingly robust to suboptimal data mixture, teacher size and compute budget. Overall, our simplified design demonstrates superior efficiency, scalability, and interpretability over the baseline V-JEPA.

## 3 EXPERIMENTAL SETUP

**Training** Our training data includes Kinetics-710 (K710) Kay et al. (2017), constructed by merging Kinetics-400/600/700 and removing all validation samples, Something-Something V2 (SSV2) Goyal et al. (2017), and a 2.8 million subset of the Panda70M Chen et al. (2024) resulting in approximately 3.6 million (3.6M) video dataset that we refer to as the V-3.6M dataset in our work. Note that our training dataset differs from the training datasets used in V-JEPA and V-JEPA 2 as the latter methods use Howto100M (Miech et al., 2019) and YT-Temporal-1B datasets (Zellers et al., 2022) while we use a subset of Panda70M in V-3.6M. Our models are standard Vision Transformers (ViT) (Dosovitskiy et al., 2020) with rotary positional embeddings (RoPE) (Su et al., 2024) which is identical to the architecture described in V-JEPA 2. Specifically, we use ViT-Large (ViT-L), ViT-Huge (ViT-H) and ViT-giant (ViT-g) in our experiments, the details of which are described in Appendix B. All baseline models (V-JEPA and V-Pixel) are trained with the same batch size of 3072 using the AdamW optimizer with hyperparameters described in detail in Appendix C. To ensure fair comparisons, we keep the number of optimization steps fixed for both our baseline methods and SALT. In other words, the total number of steps for Stage 1 and Stage 2 is identical to the number of steps used by baseline methods. The optimal number of steps to train a teacher and student is obtained via an ablation described in Section 5.4.

**Evaluation** We evaluate our models on a variety of video and image tasks. For video classification, following, we use Kinetics-400 (K400), Something-Something-v2, COIN classification (Tang et al., 2019), Jester (Materzynska et al., 2019) and Diving-48 (Li et al., 2018) freezing the backbone and training an attentive classifier to assess performance. For image classification, we adopt the same protocol on ImageNet-1K (Russakovsky et al., 2015), replicating each image 16 times to form the input sequence. Furthermore, we evaluate our models on intuitive physics understanding benchmarks, which measure performance by comparing the model's surprise scores for possible versus impossible videos. Following (Garrido et al., 2025), we use the predictor to forecast future representation. We assess performance on the IntPhys (Riochet et al., 2018), GRASP (Jassim et al., 2023), and InfLevel (Weihs et al., 2022) datasets. All setup information and hyperparameters used for our evaluations are described in detail in Appendix D.

## 4 EXPERIMENTAL RESULTS

**Systematic comparison of SALT with existing baselines** Table 1 lists the performance of SALT and existing work that serve as strong baselines including V-JEPA 2, VideoPrism (Zhao et al., 2024), InternVideo2 (Wang et al., 2024), VideoMAEv2 (Wang et al., 2023b), Perception Encoder (Bolya et al., 2025) and image encoders that include DINOv2 (Oquab et al., 2024) and SigLIP2 (Tschannen et al., 2025). We use the Kinetics-400 (K400) and Something-Something-v2 (SSv2) as benchmark datasets and evaluate SALT following the same multiclip, multiview setting used in existing baseline. We observe from Table 1 that our largest models, ViT-g and ViT-G, trained with SALT outperforms all of the baseline methods on SSv2, which tests the motion understanding ability of video models. On K400, which is an appearance understanding benchmark, the encoders trained with our method

---

[2]This method is implicitly described in Table 1 by Bardes et al. (2024). We name the method V-Pixel for clarity in presentation.

Table 1: **Systematic comparison** of state-of-the-art video encoders under frozen-backbone evaluation, using SSv2 (16×2×3) and K400 (16×2×3). The comparison includes several baselines including encoders trained with V-JEPA 2 method on our V-3.6M dataset. The V-JEPA 2 encoders trained on V3.6M and SALT encoders are evaluated using the protocol in Section 3 and Appendix D. The results for other models are duplicated from Table 4 in V-JEPA (Assran et al., 2025). A detailed description of FLOPs calculation is available in Appendix F.

| Method | Param. | Pretraining Dataset | Teacher (Params) | Total Compute | # Seen Samples | SSv2 | K400 |
|---|---|---|---|---|---|---|---|
| VideoMAEv2 (Tong et al., 2022) | 1B | UnlabeledHybrid-1.4M | N/A | 2.2 | 1.6B | 56.1 | 82.8 |
| MVD (Wang et al., 2023c) | 300M | IN1K+K400 | N/A | — | — | 66.5 | 79.4 |
| PE$_{core}$G (Bolya et al., 2025) | 1.9B | MetaCLIP-5B (Xu et al., 2024) | N/A | — | 86B | 55.4 | 88.5 |
| InternVideo2-1B (Wang et al., 2024) | 1B | IV-25.5M | InternVL-6B + VideoMAEv2-g (6B + 1.0B) | — | — | 67.3 | 87.9 |
| VideoPrism (Zhao et al., 2024) | 1B | VT-36M | Stage-1-ViT-g ( 1.0B) | — | 2.0B | 68.5 | 87.6 |
| DINOv2 (Oquab et al., 2024) | 1.1B | LVD-142M | EMA teacher (1.1B) | — | 1.9B | 50.7 | 83.6 |
| SigLIP2 (Tschannen et al., 2025) | 1.2B | WebLI-10B (Chen et al., 2022) | EMA teacher (1.2B) | — | 40B | 49.9 | 87.3 |
| V-JEPA 2 ViT-L (Assran et al., 2025) | 300M | VM-22M | EMA teacher (300M) | 1.9 | 0.7B | 73.7 | 85.1 |
| V-JEPA 2 ViT-H (Assran et al., 2025) | 600M | VM-22M | EMA teacher (600M) | 3.5 | 0.7B | 74.0 | 85.3 |
| V-JEPA 2 ViT-g (Assran et al., 2025) | 1B | VM-22M | EMA teacher (1B) | 5.3 | 0.7B | 75.3 | 86.6 |
| V-JEPA 2 ViT-L | 300M | V-3.6M | EMA teacher (300M) | 1.4 | 0.7B | 68.2 | 83.8 |
| V-JEPA 2 ViT-H | 600M | V-3.6M | EMA teacher (600M) | 2.6 | 0.7B | 73.4 | 84.6 |
| SALT ViT-L | 300M | V-3.6M | | 1.2 | 0.7B | 74.9 | 85.4 |
| SALT ViT-H | 600M | V-3.6M | SALT-ViT-L (300M) | 1.5 | 0.7B | 75.4 | 86.0 |
| SALT ViT-g | 1B | V-3.6M | | 1.9 | 0.7B | 76.2 | 86.8 |
| SALT ViT-G | 2B | V-3.6M | | 2.6 | 0.7B | 76.1 | 87.2 |

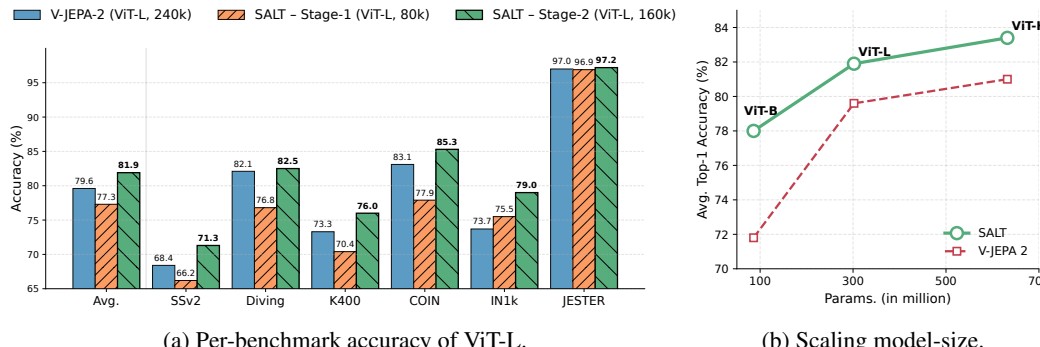

(a) Per-benchmark accuracy of ViT-L.          (b) Scaling model-size.

Figure 2: **V-JEPA 2 vs. SALT at matched total steps on V-3.6M.** Both methods are trained on the same V-3.6M dataset for an identical number of pretraining steps. SALT uses an 80k-step teacher and a 160k-step student. We evaluate all models under the *same frozen-backbone protocol* across standard video/image benchmarks: K400 with 16×1×1 input; SSv2 with 16×1×1; Diving48 and Jester with 16×4×3; and COIN 16×8×3. Table 12 provides a breakdown of downstream performance for each dataset used in this evaluation.

exceeds the performance of V-JEPA 2 across all scales and remains highly competitive with other state-of-the-art methods including the recently proposed Perception Encoder.

**Static teacher improves representation quality** A key design choice of SALT is the use of a *static* teacher which differs from the dynamic momentum-encoder teacher used in V-JEPA 2. In order to ascertain the differences between these two approaches, we use the same V-3.6M dataset and same input resolution of 224 × 224 to train SALT and V-JEPA 2. We train both methods for a total of 240k steps in this study. Figure 2a shows the downstream performance results of this study for ViT-L-based teacher and student model setup while Figure 2b shows the scaling behavior of the two methods as we scale up the student encoder while using a teacher encoder that is the same size or smaller than the student. We observe from Figure 2a that SALT improves the average accuracy over the V-JEPA 2-based encoder by 2.3% where the average is calculated over six benchmarks. Furthermore, we observe from Figure 2b that SALT displays improved performance as we scale up the student. Note that we use the same-sized teacher student for ViT-B and ViT-L while we use a smaller ViT-L teacher model for training ViT-H student encoder. We refer the reader to Appendix B for detailed model size and other architecture information. Together, Figure 2 suggests that the *static* teacher-based SALT learns higher quality features when compared to V-JEPA 2 that uses an EMA-based teacher.

**Small teachers unlock compute efficiency** Table 1 and fig. 2 show that strong students can be trained from a *frozen* teacher, which is considerably cheaper: a fixed ViT-L teacher success-

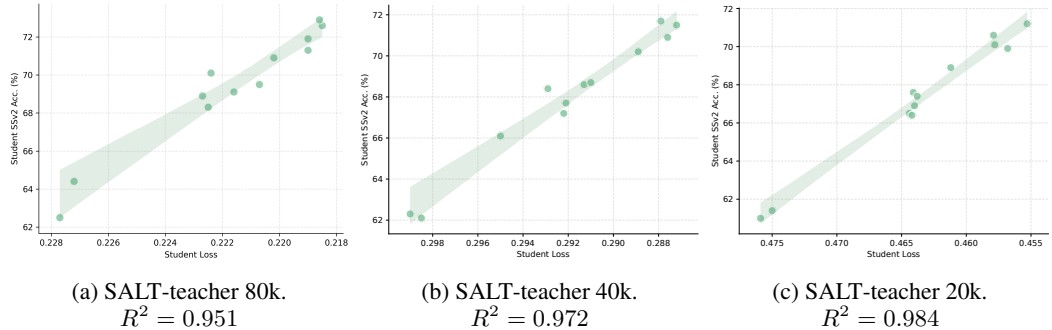

(a) SALT-teacher 80k.
$R^2 = 0.951$

(b) SALT-teacher 40k.
$R^2 = 0.972$

(c) SALT-teacher 20k.
$R^2 = 0.984$

Figure 3: **Correlation between student loss and downstream accuracy.** Observe that the student model's training loss is predictive of downstream accuracy.

fully trains same-size ViT-L students, and much larger ViT-H/g/G students. Consequently, SALT achieves lower *total* pretraining FLOPs than the EMA-based baseline across model sizes, even when accounting for the teacher pretraining stage. The savings stem from the simple, efficient teacher pretraining (e.g., ViT-L on V-3.6M) and grow with both model size and spatial resolution. FLOPs computation details appear in Appendix F.

**SALT enables interpretable model selection** A key challenge with using joint-embedding methods including JEPA is that the training loss is typically uninformative of representation quality. Figure 3a shows student training loss versus student downstream accuracy plot for various student checkpoints that use the same SALT Stage 1 checkpoint as the teacher. This checkpoint is obtained by training a teacher for 80K steps. The plot shows that the student loss is highly predictive of the downstream accuracy with an $R^2$ value of 0.951, suggesting an almost linear relationship. This result implies that SALT significantly simplifies tracking the quality of representation during student pretraining, and provides a clear signal for improvement via simple loss minimization. Similar observations can be made by a teacher trained for 40K and 20K iterations in Figure 3. Lastly, we study whether teacher-related metrics such as teacher loss or RankMe (Garrido et al., 2023) are predictive of downstream performance in Figure 9. We find that neither the teacher's loss nor embedding rank are predictive of the student encoder's downstream performance.

**Intuitive physics evaluation** Garrido et al. (2025) have shown that video models trained with the JEPA objective show an emergent understanding of intuitive physics. We follow the setup described in (Garrido et al., 2025) to measure the intuitive understanding ability of video models trained with SALT. The evaluation setup and results are discussed in detail in Appendix E.1 due to space limitations. The main finding is that intuitive physics understanding is observed on models trained via SALT as well as V-JEPA 2.

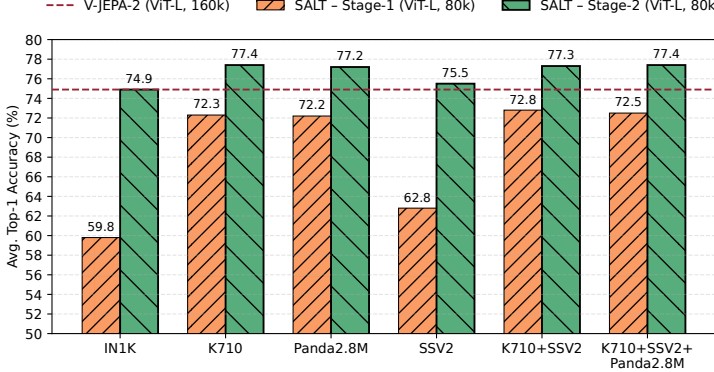

Figure 4: **Training data of static-teacher.** We ablate the impact of training data of teacher, thus fixed the student's training data as the whole data-mix by default. Table 13 provides a detailed breakdown of the results show above.

## 5 TEACHER DESIGN CHOICE ABLATION

The empirical analysis in Section 4 shows that SALT provides high-quality representation that outperforms several existing methods. A key aspect of SALT is the static (frozen) teacher that provides high-quality prediction targets. We study the design choices involved in training the teacher model and student model that lead to optimal representation via a series of ablations described in the following.

### 5.1 TRAINING DATASET

In this section, we study the role of pretraining data distribution on a teacher model. Specifically, we train a ViT-L teacher model with six training datasets: (i) Kinetics-710 (K710), (ii) Something-Something-V2 (SSv2), (iii) a 2.8 million subset of Panda70M, (iv) ImageNet-1k, (v) data aggregated from K710 and SSv2, and (vi) V-3.6M which is data aggregated from K710, SSv2 and Panda70M subset. The exact details of the datasets are described in Section 3. The teacher model is trained using the Stage 1 approach (V-Pixel) described in Section 2.2. For each teacher model from Stage 1, we train a ViT-L-based student model on the combined V-3.6M dataset using Stage 2 approach described in Section 2.2.

Figure 4 shows the result of benchmarking the teacher and student models trained with the datasets described above. We observe that the performance of each student model described above improves over that of its corresponding teacher. Additionally, each student model's downstream performance exceeds the performance of a V-JEPA 2-based encoder with the exception of the student model trained on ImagetNet-1K-based teacher model. Among the teacher models trained on video datasets considered in this study, we observe comparable performance with the notable exception of the teacher model trained on Something-Something-V2. Taken together, these results suggest that an effective teacher may be trained with a relatively small amount of data to build strong foundation models.

### 5.2 TEACHER MASKING STRATEGY

We study the role of masking strategy used to train the teacher that provides targets to optimize the student model. To this end, we train a ViT-L using random masking used in Video-MAE, multi-block masking used in V-JEPA and a modified method that we call multi-random tube where we adapt the short-range and long-range masking idea from V-JEPA to random masking. We refer the reader to Table 15 for setup details used in this ablation. Figure 5 shows the results for this ablation. We observe that the multi-block masking approach works the best for V-Pixel model achieving an accuracy of 72.5%. This result in and of itself is a new empirical finding as VideoMAE models typically use random-tube masking. Furthermore, we observe from Figure 5 that the student trained with a multi-block teacher achieves the

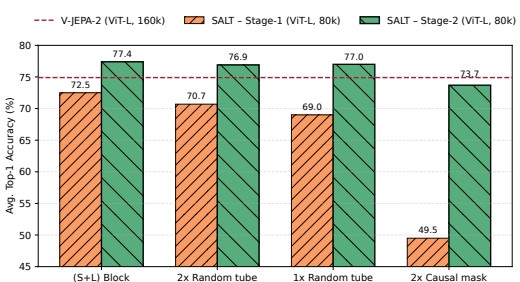

Figure 5: **Masking Strategy of static-teacher.** We study the impact of random vs multi-block masking strategy influences a student's performance. Table 15 includes hyperparameters and results information.

highest accuracy while the other student models also show a big improvement over their corresponding teachers. We conclude that multi-block masking strategy is effective with training our teacher and name the pixel reconstruction method with multi-block masking as V-Pixel.

### 5.3 TEACHER MODEL SIZE

Next, we study the impact of a teacher model's size on a student model's performance. We train a ViT-B, ViT-L, ViT-H and ViT-G based V-Pixel models and use these models to supervise ViT-L and ViT-G based student models. Figure 6 shows the results of this ablation. We observe that the best performing ViT-L student has an average accuracy of 77.4% and is obtained by training with a

ViT-L teacher. This result is remarkable as this accuracy is better than the accuracy obtained with ViT-H and ViT-G based teachers that are larger than the student. A similar observation can be made about the ViT-G student where the highest average accuracy of 78% is obtained with a ViT-L teacher. Additionally we observe that all student models show improvement over their teachers which are of the same or smaller size. These observations suggest that the multi-stage training proposed in SALT allow the student to bootstrap from a weaker teacher to learn high-quality representation.

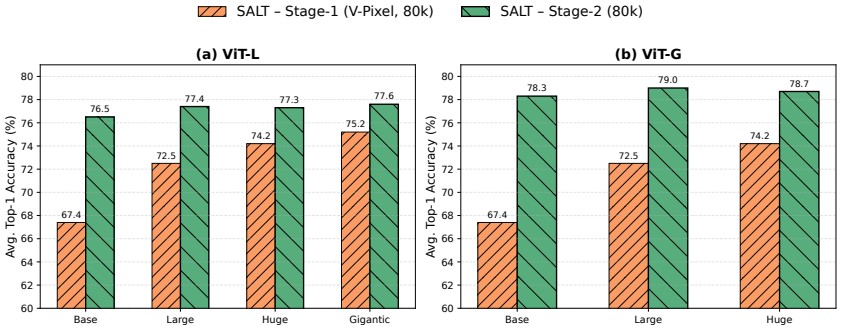

Figure 6: **Teacher model size ablation.** We train ViT-B, ViT-L, ViT-H and ViT-G based teacher and use the teacher to train a ViT-L and ViT-G student. Observe that the best performing student is obtained from ViT-L teacher with modest performance. Table 14 provides a detailed breakdown of results on downstream benchmarks.

## 5.4 IMPACT OF TEACHER-STUDENT COMPUTE ALLOCATION

Due to a frozen teacher approach used in SALT, we are confronted with the problem of allocating compute between the teacher and the student. Training compute is a function of the model size, the number of optimization steps as well as the number of FLOPs per step. In this ablation, we hold the model size, the total number of optimization steps, and the training dataset (V-3.6M) constant as that allows us to conduct a fair comparison of SALT and V-JEPA 2 baseline.

We use ViT-L based model for both teacher and student in this ablation where we vary the number of steps used to train the teacher and student for a fixed total number of optimization steps. We observe from Figure 7 that SALT outperforms V-JEPA 2 baseline over a range of optimization steps considered in our experiments. Furthermore, we observe that SALT exceeds V-JEPA 2's performance at ≈ the same FLOPs level which in turn suggests that SALT is more compute efficient than V-JEPA 2. The best performing student is obtained by training on 240k total steps, as can be seen from Figure 7, that is supervised by a teacher that is only trained for 40k steps. This finding underscores the effectiveness of our multi-stage training approach and demonstrates that we should focus on students in SALT, and that aiming for a high-performing teacher may be wasteful. Additional visualization and analysis that supports these claims are provided in Appendix E.2.

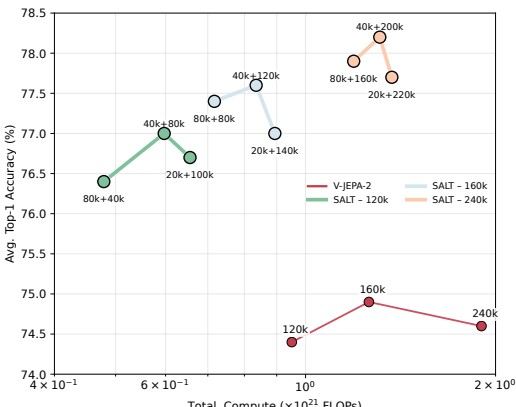

Figure 7: **Comparison of compute allocation in SALT.** We show average Top-1 accuracy across benchmarks against total training FLOPs. Our SALT curves dominate V-JEPA-2 at matched budgets. See Table 16 for additional details.

## 6 RELATED WORK

**Video foundation models:** Masking-based self-supervised learning (SSL) (Tong et al., 2022; Feichtenhofer et al., 2022; Wang et al., 2023a; Bardes et al., 2024; Assran et al., 2025; Wang et al., 2023c; Li et al., 2023; Zhao et al., 2024; Wang et al., 2022b) is a prominent approach used to learn representation from large-scale video datasets for building video foundation models. Several

works (Tong et al., 2022; Feichtenhofer et al., 2022; Wang et al., 2023b) have extended image-based masked autoencoders (He et al., 2021) to video data by using random masking to learn representation via pixel-space reconstruction. Recent works incorporate motion-aware masking and temporal correspondence to better capture temporal dynamics (Thoker et al., 2025; Sun et al., 2023; Salehi et al., 2024; Huang et al., 2023). An alternate approach for representation learning is to learn via latent-space predictions. These methods are known to learn features that differ in quality from those obtained via reconstruction-based methods (Littwin et al., 2024; Balestriero & Lecun, 2024). Prominent among latent-space prediction methods for video are V-JEPA (Bardes et al., 2024) and V-JEPA 2 (Assran et al., 2025) that use an online/momentum encoder to learn the teacher that provides prediction targets. SALT simplifies the JEPA pipeline by using a frozen teacher that is reliable and efficient that leads to higher quality representation as shown in our analysis.

**Distillation from frozen teacher encoder:**   While many studies utilize frozen pretrained models as teachers for student encoder supervision, we limit our review to prior work that is directly relevant to our core method and refer the interested reader to (Balestriero et al., 2023) for a comprehensive survey on SSL literature. MVD (Wang et al., 2023c) and InternVideo (Wang et al., 2022b) use VideoMAE (Tong et al., 2022) while other works such as UnMasked Teacher (Li et al., 2023), VideoPrism Zhao et al. (2024), InternVideo2 Wang et al. (2024) and many more use a vision-language model (Radford et al., 2021), as a frozen teacher. PerceptionEncoder (Bolya et al., 2025) is a recent vision foundation model that uses features from a predefined layer from within the model as well as features from an external teacher SAM (Kirillov et al., 2023) to encourage feature locality. AM-RADIO (Ranzinger et al., 2024) proposes to learn a student from multiple teacher models. Beyond prior work, which require access to powerful pretrained encoders, we uncover a *weak-teacher, strong-student* effect: students supervised by much weaker *frozen* teachers consistently outperform those trained with EMA-based teachers. Our method is purely self-supervised and unregularized, unlike self-training Xie et al. (2020), which relies on labeled and unlabeled data and explicit noise regularization. Related world-modeling approaches (Karypidis et al., 2024; Baldassarre et al., 2025b; Zhou et al., 2024) also fix encoders for stability but optimize for future-state prediction, while our aim is representation learning. Nevertheless, SALT 's strong video features make it a promising backbone for world models. Lastly, recent advancements in image-to-video distillation (Li & Liu, 2023; Hu et al., 2022; Liu et al., 2025) incorporate temporal information into strong representation models during distillation while we learn representations directly from video.

**Masked video distillation:**   Wang et al. (2023c) propose a two-stage method called masked video distillation (MVD) that first trains two separate encoders one each for image and video input using an MAE (He et al., 2021; Tong et al., 2022)-like approach. These teacher encoders then provide targets (latent features) used to optimize a smaller student encoder. SALT resembles the approach taken by MVD but has several critical differences. The first difference is that we provide an improved method to train the video teacher encoder as a result of careful empirical analysis in Figure 5. Additionally, we do not use a separate encoder for image data but instead focus on learning representation from large-scale video datasets. The most critical difference is that we use a teacher model whose size is the same or smaller than that of the student model. Furthermore, we conduct detailed ablations in Section 5 to show how to choose a teacher model (checkpoint). Finally, SALT learns superior features as our benchmark results are based on frozen backbone evaluations while MVD uses fine-tuning for downstream evaluation.

## 7   LIMITATIONS AND CONCLUSION

We present SALT, a simple, compute-efficient, and scalable framework for video representation learning. Across standard benchmarks, SALT consistently outperforms strong baselines, including V-JEPA-2 in frozen-evaluation protocols. Strikingly, we find that *sub-optimal*, often smaller teachers can yield much stronger students, raising questions as to how the quality of the teacher should be assessed, and whether EMA-based machinery is necessary to learn highly-semantic representation. Prior works including SimSiam (Chen & He, 2021) and Asym-Siam (Wang et al., 2022a) have explored the hypothesis high quality student encoders can be trained using targets from a teacher encoder that provide lower variance targets. Exploring the applicability of this hypothesis to our setup as well as building a principled characterization of teacher quality is left as future theoretical work. On the empirical front, a fuller study of SALT 's scaling behavior with respect to data and

model size is left as future work.

While SALT improves compute efficiency and downstream performance over self-distillation, it has limitations. Our ablations (Section 5) suggest that a simple *V-Pixel* recipe usually suffices to train an effective teacher and that compute is best allocated to the student; however, they do not fully explain what makes a "good" teacher. We also observe modest gains from an additional student-training stage, but the mechanism remains unclear. Given the experiment volume, we focused compute on simple scalar diagnostics of teacher quality (Section 5 and Appendix E.3). Finally, performance plateaus as model size grows in our setting, likely reflecting data limits, and that larger pretraining sets may extend the scaling trend.

ACKNOWLEDGMENTS

We thank Eeshan Gunesh Dhekane and Preetum Nakkiran for their insightful feedback and discussions provided during the process of preparing this manuscript. VT would like to thank Meng Cao for advice on video datasets. We thank the entire Apple Machine Learning Research team for providing resources and inspiration that allowed us to complete this project.

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

## A    TRAINING DATASET

We describe the datasets used to train vision transformer (ViT) models with SALT. We form the Kinetics-710 (K710) dataset by combining training samples from Kinetics-400/600/700 (Kay et al., 2017) and removing duplicated samples as well as samples that are in the validation sets of the above datasets. We then add training samples from the Something-Something-v2 (SSv2) (Goyal et al., 2017) dataset. Finally, we add an approximately 2.8 million video clips subset of Panda70M (Chen et al., 2024) to form our dataset that we refer to as V-3.6M to train our models. We apply stratified sampling to select the subset of clips from Panda70M that enables us to have clips whose duration ranges from 4 seconds to 50 seconds. We do not apply any other form of filtering or curation to form our training dataset. Table 2 lists the sample count information for our V-3.6M dataset.

Table 2: **V-3.6M** Training dataset details.

| Dataset | Sample Count |
|---|---|
| Kinetics-710 | 657,257 |
| Something-Something-v2 | 168,913 |
| Panda70M | 2,799,959 |
| **V-3.6M** | 3,626,129 |

## B    ARCHITECTURE DETAILS

We use Vision Transformers (ViTs) (Dosovitskiy et al., 2020) to implement our video encoders and predictors. We use a spatial patch size of $16 \times 16$ and temporal patch size of 2 in all of our models. Table 3 and Table 4 lists the model architecture in detail for our encoders and predictors respectively. We follow V-JEPA 2 (Assran et al., 2025) and use rotary position embedding (RoPE) (Su et al., 2024) to encode position information in all of our models. Note that our predictor's last layer projects the embedding dimension to be compatible with that of the teacher encoder. This information is captured in the input and output dimension columns in Table 4. We use a ViT-L teacher to train all encoders except the ViT-B model which uses a ViT-B teacher. This information is captured in the output dimension column in Table 4.

Table 3: Encoder model architecture details. M indicates a million and B a billion.

| Model | Parameter Count | Width | Depth | Heads |
|---|---|---|---|---|
| ViT-B | 86M | 768 | 12 | 12 |
| ViT-L | 303M | 1024 | 24 | 16 |
| ViT-H | 632M | 1280 | 32 | 16 |
| ViT-g | 1.012B | 1408 | 40 | 16 |
| ViT-G | 1.843B | 1664 | 48 | 16 |

Table 4: Predictor model architecture details. M indicates a million.

| Predictor & (Encoder) | Input Dimension | Output Dimension | Parameter Count | Width | Depth | Heads |
|---|---|---|---|---|---|---|
| ViT-Predictor (ViT-B) | 768 | 768 | 21.88M | 384 | 12 | 16 |
| ViT-Predictor (ViT-L) | 1024 | 1024 | 22.08M | 384 | 12 | 16 |
| ViT-Predictor (ViT-H) | 1280 | 1024 | 22.18M | 384 | 12 | 16 |
| ViT-Predictor (ViT-g) | 1408 | 1024 | 22.23M | 384 | 12 | 16 |
| ViT-Predictor (ViT-G) | 1664 | 1024 | 22.32M | 384 | 12 | 16 |

Table 5: Hyperparameter details used to train models with SALT. Note that "indicates that Stage 2 uses the same hyperparameter value as listed in Stage 1.

| Parameter | Stage 1 | Stage 2 |
|---|---|---|
| Input spatial resolution | $224 \times 224$ | " |
| Tubelet size | 2 | " |
| Patch size | $16 \times 16 \times 2$ | " |
| Number of frames | 16 | " |
| Frame step | 4 | " |
| Random resize aspect ratio | [0.75, 1.35] | " |
| Random resize scale | [0.3, 1] | " |
| Short-range Spatial mask scale | 0.15 | " |
| Long-range Spatial mask scale | 0.7 | " |
| Temporal mask scale | 1 | " |
| Mask aspect ratio | [0.75, 1.5] | " |
| Batch size | 3072 | " |
| Number of Steps | Variable | Variable |
| Steps per epoch scale | 1 | " |
| Start learning rate | 0.0002 | " |
| learning rate | 0.000625 | " |
| Final learning rate | 1e-6 | " |
| Start Weight decay | 0.04 | " |
| End Weight decay | 0.4 | " |
| Clip grad | 0.02 | " |
| Learning rate schedule | Cosine | " |
| Warmup steps | 10000 | " |
| AdamW $\beta_1$ | 0.9 | " |
| AdamW $\beta_2$ | 0.95 | " |

## C  TRAINING DETAILS

Recall from Section 2.2 SALT is a multi-stage training approach in which the teacher is trained at first via V-Pixel method followed by student training using a frozen or static teacher in the last stage. Table 5 lists hyperparameter information in detail that are used to train video encoders with SALT. Observe that we use multi-block masking method (Bardes et al., 2024) for V-Pixel. Note that the hyperparameters for setting up multi-block are copied over from those used in V-JEPA (Bardes et al., 2024).

Table 5 also lists optimization-related hyperparameters that we used to train video encoders. We use a value of $240,000$ steps in total to show results in Table 1. We conduct ablations with the number of steps set to $120,000$, $160,000$, or $240,000$ for results shown in Figure 2 and discussed in Section 5. Observe that we use the standard cosine weight-decay strategy during training (Bardes et al., 2024). We use a value of 0.95 for $\beta_2$ in AdamW (Loshchilov & Hutter, 2017) as this value is used by Carreira et al. (2024) to train VideoMAE-like models both at scale but most importantly at large scale. We also opt not to use virtual early stopping approach adopted in V-JEPA (Bardes et al., 2024) and V-JEPA 2 (Assran et al., 2025) that scales the training steps by $25\%$ to avoid training instabilities in the latter part of training. Empirically, we observe that frozen teacher provides a stable representation that allows SALT to be stable throughout training.

## D  EVALUATION DETAILS

We adopt the evaluation protocol used in V-JEPA 2 (Assran et al., 2025) that uses attentive probing to ensure fair comparison between SALT, V-JEPA 2 and several baselines reported by Assran et al. (2025). We use Kinetics-400 (Kay et al., 2017), Something-Something-v2 (SSv2) (Goyal et al., 2017) to systematically compare against state-of-the-art baselines the results of which are reported in Table 1.

**Systematic evaluation setup for K400 and SSv2** We use inputs with 16 frames, 8 segments or clips per input and 3 spatial views per segment which is identical to the setting used in V-JEPA 2 (Assran et al., 2025) for this dataset. The probe consists of attentive pooling which is implemented via four Transformer blocks where the first three blocks are self-attention based blocks while the last layer uses cross-attention with a learnable query token. This pooling operation is followed by a standard linear layer where the number of outputs is set to the number of classes for a classification dataset. This value is 400 for Kinetics-400 dataset and 174 for SSv2 dataset.

The attentive probe is trained with AdamW for 20 epochs using a learning and weight decay hyperparameters that are determined via a grid search. Table 6 reports the hyperparameters that are common to SALT and V-JEPA 2.

The key difference between SALT and V-JEPA 2 is that we use a spatial crop of $224 \times 224$ while V-JEPA 2 uses $256 \times 256$. This difference makes the results obtained with SALT even more remarkable as we spend much less compute during probing compared to V-JEPA 2 due to using smaller resolution.

Table 6: Kinetics-400 and Something-Something-v2 evaluation hyperparameters that are common to SALT and V-JEPA 2. The results of this evaluation are shown in Table 1. Note that "denotes the value is the same as the one used in K400 evaluation.

| Parameter | K400 | SSv2 |
|---|---|---|
| Number of frames | 16 | " |
| Segments / Clip | 8 | 2 |
| Views / Segment | 3 | " |
| Frame step | 4 | " |
| Epochs | 20 | " |
| Batch size (global) | 256 | " |
| Classifier heads | 20 | " |
| Classifier learning rates | [5e-3, 3e-3, 1e-3, 3e-4, 1e-4] | " |
| Classifier weight decay | [.8, .4, .1, .01] | " |

**Fast evaluation setup for K400 and SSv2** Due to the sheer volume of compute involved with training and probing our methods over a range of downstream datasets, we use a more efficient evaluation protocol for many ablations and results shown in the main paper. The main difference is the use of 1 clip and 1 view per frame while keeping the 16 frames per input clip as described above. The evaluation hyperparameters identical to the values used in V-JEPA (Bardes et al., 2024) and are described in Table 7 for completeness. The results of these evaluations are described in Figures 1 to 7, 8 and 9.

Table 7: Kinetics-400 and Something-Something-v2 evaluation hyperparameters that are common to SALT and V-JEPA 2.

| Parameter | K400 & SSv2 |
|---|---|
| Number of frames | 16 |
| Segments / Clip | 1 |
| Views / Segment | 1 |
| Frame step | 4 |
| Epochs | 20 |
| Batch size (global) | 256 |
| Classifier heads | 1 |
| Classifier learning rates | 1e-3 |
| Classifier weight decay | .01 |

**COIN, Diving-48 and Jester Evaluations** We report results using COIN classification (Tang et al., 2019), Diving-48 (Li et al., 2018), Jester (Materzynska et al., 2019) and ImageNet-1K (Russakovsky et al., 2015) benchmarks in addition to Kinetics-400 and Something-Something-v2 bench-

marks. The number of classes in COIN, Diving-48, Jester, and ImageNet-1K are 180, 48, 27 and 1000 respectively. Table 8 reports the hyperparameters used to evaluate frozen backbones with these benchmarks. The results of these evaluations are reported in Figures 1 to 7, 8 and 9.

Table 8: COIN (Tang et al., 2019), Jester (Materzynska et al., 2019), Diving-48 (Li et al., 2018) and ImageNet-1K (Russakovsky et al., 2015) evaluation hyperparameters that are common to SALT and V-JEPA 2. Note that **"** denotes the value is the same as the one used in COIN evaluation.

| Parameter | COIN | Jester/Diving-48 | ImageNet-1K |
|---|---|---|---|
| Number of frames | 16 | " | " |
| Segments / Clip | 8 | 4 | 1 |
| Views / Segment | 3 | " | " |
| Frame step | 4 | 2 | NA |
| Epochs | 20 | " | " |
| Batch size (global) | 256 | 128 | 1024 |
| Classifier heads | 1 | " | " |
| Classifier learning rates | 1e-3 | " | " |
| Classifier weight decay | .01 | " | " |

**Intuitive physics** We follow the protocol established by Garrido et al. (2025) and use the surprise score in our evaluations with IntPhys-2019 or IntPhys (Riochet et al., 2018), GRASP (Jassim et al., 2023) and InfLevel (Weihs et al., 2022) datasets. In the following, we reproduce the equations used by Garrido et al. (2025) to quantify surprise. We let $f$ be the context encoder, $g$ be the predictor or the decoder and $h$ be the target encoder. $V$ denotes a frame of a video clip, $C$ denotes the context frame count and $M$ denotes the number of future frames. The surprise at time $t$ is given by:

$$S_t = \|g_\phi \left( f_\theta \left( V_{t:t+C} \right) \right) - h_\psi \left( V_{t:t+C+M} \right) \|_1. \tag{2}$$

The surprise above can then be calculated over all windows to obtain the following **global surprise score**:

$$\text{Average Surprise} = \frac{1}{T} \sum_{t \in \{1, 1+s, \dots, T-(C+M)\}} S_t \quad \text{or} \quad \text{Maximum Surprise} = \max_{t \in \{1, 1+s, \dots, T-(C+M)\}} S_t. \tag{3}$$

,

where we set s to 2 and use the average surprise score to quantify the surprise between a pair of videos following the methodology used by Garrido et al. (2025). The scores are then converted to relative accuracy using label information for video pairs to obtain the relative accuracy values discussed in Table 9.

# E  ADDITIONAL RESULTS

In this section, we include additional tables and results to support figures and tables in the main paper.

## E.1  INTUITIVE PHYSICS BENCHMARKS

In this section, we evaluate the intuitive physics understanding of video models. We follow the protocol and datasets described in Garrido et al. (2025) to test a video model's understanding of intuitive physics in a zero-shot setting. Following the protocol of Garrido et al. (2025) we calculate a surprise metric that measures the deviations from expected physical behavior. The benchmark probes the predictor or the decoder to test for physical attributes such as object permanence, spatio-temporal continuity, shape and color constancy, gravity, support, solidity, inertia and collision. We refer the interested reader to Garrido et al. (2025) for additional details on datasets and definition of the attributes mentioned above.

Table 9: Comparison on Intuitive physics benchmarks (IntPhys, GRASP, InfLevel).

| Method | Encoder | Predictor | IntPhys | GRASP | InfLevel | Avg |
|---|---|---|---|---|---|---|
| *Results reported in (Baldassarre et al., 2025a)* | | | | | | |
| COSMOS-4B (Agarwal et al., 2025) | VAE | 4B | 99.5 | 60.1 | 44.8 | 68.1 |
| V-JEPA (Bardes et al., 2024) | ViT-L | 22M | 92.2 | 67.0 | 58.9 | 72.7 |
| V-JEPA (Bardes et al., 2024) | ViT-H | 22M | 89.4 | 73.0 | 59.9 | 74.1 |
| *Results reported in (Bordes et al., 2025)* | | | | | | |
| V-JEPA 2 Assran et al. (2025) | ViT-H | 22M | 87.2 | – | – | – |
| *Our Eval* | | | | | | |
| VideoMAE V2 (Wang et al., 2023b) | ViT-g | 12M | 59.4 | 61.3 | 54.4 | 58.4 |
| V-JEPA 2 (V-3.6M) | ViT-B | 22M | 76.6 | 56.1 | 52.4 | 61.7 |
| V-JEPA 2 (V-3.6M) | ViT-L | 22M | 96.9 | 53.0 | 57.4 | 69.1 |
| V-JEPA 2 (V-3.6M) | ViT-H | 22M | 88.5 | 65.1 | 60.0 | 71.2 |
| SALT | ViT-B | 22M | 72.4 | 56.4 | 54.9 | 61.2 |
| SALT | ViT-L | 22M | 90.6 | 53.5 | 54.2 | 66.1 |
| SALT | ViT-H | 22M | 95.8 | 58 | 58.2 | 70.7 |

Table 9 shows the results of this benchmark for SALT as well as several baseline video models. Table 9 shows that SALT's average accuracy scales with model size. SALT compares favorably with published baselines including COSMOS-4B (Agarwal et al., 2025), V-JEPA (Bardes et al., 2024) and V-JEPA 2 (Assran et al., 2025) and VideoMAEv2 (Wang et al., 2023b). Finally, we consider a setup where we train V-JEPA 2 and SALT models using the same dataset and optimization budget. We observe that V-JEPA 2 models trained under the conditions stated above compare favorably with SALT. These results suggests that emergent intuitive physics understanding behavior observed in video models trained with V-JEPA objective (Garrido et al., 2025) is seen with SALT as well.

## E.2 IMPACT OF TEACHER-STUDENT COMPUTE ALLOCATION

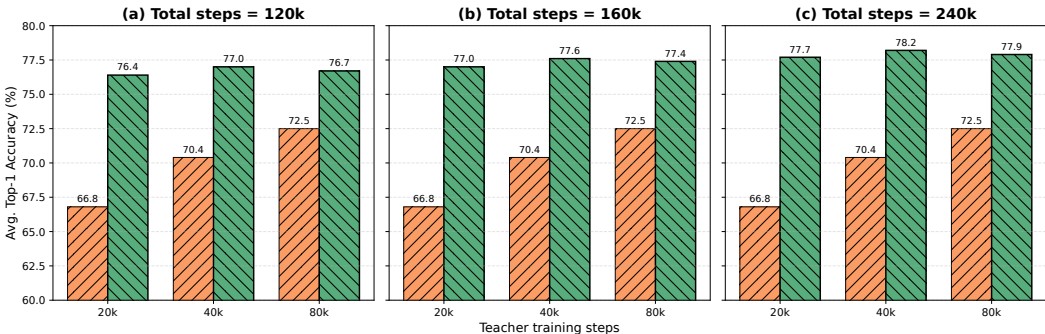

Figure 8: SALT trained with a compute budget of (a) 120K steps (b) 160K steps and (c) 240K steps. The X-axis shows the number of steps allocated to the teacher with the rest used to optimize the student. Observe that the optimal allocation favors training the student longer than the teacher.

Figure 8 provides an alternative view of the plot shown in Figure 7. We train a ViT-L model in this ablation. It is clear from Figure 8 that the teacher encoder's downstream performance increases with the number of training steps across all values of total number of training steps. Remarkably, the student encoders improve over the teachers that they use to obtain predictions targets. The best performing model is obtained by training a teacher for 40,000 steps and using the remaining steps on the student. This observation suggests that the optimal compute allocation should favor the student.

## E.3 HOW TO CHOOSE A TEACHER CHECKPOINT?

A question that arises naturally with SALT is whether there is a principled way to choose an optimal teacher checkpoint. By optimal, we here mean choosing a checkpoint that maximizes the student's performance. We study this question empirically by looking at the correlation between the student's benchmark accuracy and teacher's embedding rank, training loss and teacher model's downstream accuracy. Figure 9a shows a plot of embeddings rank where the embeddings are extracted from a teacher model versus student accuracy. We use RankMe (Garrido et al., 2023) to estimate the embedding rank. Garrido et al. (2023) have shown that high embedding rank is a necessary condition

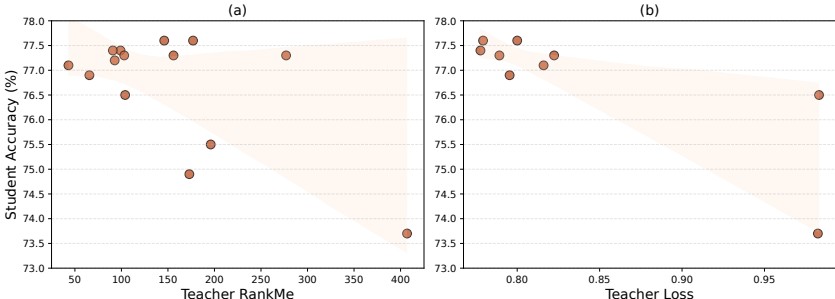

Figure 9: Teacher quality vs. student performance. We take all the teachers trained in SALT and measure the RankME(Garrido et al., 2023) of the embedding and pretraining loss and analyze the correlation between them and student's downstream performance. Each point represents a single SALT run. We control the total training budgets of 160k for both stages to the be same for all models in this comparison.

for good downstream performance in joint-embedding self-supervised learning (JE-SSL) models. We observe from Figure 9 that the the teacher's embedding rank is not predictive of student's downstream performance. A similar trend can be observed from Figure 9b with the teacher's pixel reconstruction or training loss. We see that neither the teacher loss nor its embedding rank are predictive of downstream student's performance.

# F  FLOATING POINT OPERATIONS (FLOPS) ESTIMATION

A common approach to estimating the total training compute for Transformers, including ViTs, is by using the well-known $6ND$ formula (Kaplan et al., 2020; Hoffmann et al., 2022). Here $N$ stands for the model parameter count while $D$ represents the total number of tokens used to train the model. This simple approximation assumes that the backward pass during training costs twice as much as the forward pass. Consequently, we use $2ND$ to approximate the training compute of a teacher model that provides targets in distillation-based methods. The total number of input tokens observed by a model during training is a function of input resolution, spatial for image models and spatio-temporal for video models, the patch size, the batch size per step and the total number of optimization steps. Note that we include the embedding layer in our parameter count. With these preliminaries in place, we present the total compute estimate for models presented in Table 1.

**VideoMAEv2 (Wang et al., 2023b)**  We use a masking ratio of 0.9 and 0.5 for Video-MAEv2 (Wang et al., 2023b) encoder and decoder respectively. The model considered here is a ViT-g model with an input patch size of 14 that operates on a spatial input of size $16 \times 16$. Our analysis uses 1200 epochs for calculating the number of tokens in the encoder and decoder. The other details used to estimate FLOPs is shown in Table 10.

**V-JEPA 2 (Assran et al., 2025)**  The spatial resolution is $256 \times 256$ with a patch size of 16. We use a masking ratio of 0.9 for the encoder on average following the recommendation made in V-JEPA (Bardes et al., 2024). The predictor operates on a token count that is half of that seen by the encoder due to temporal stride being set to 2. In other words, the predictor sees the union of mask tokens used for missing regions and context tokens used for visible regions. We assume that the model is trained for 240,000 steps using a batch size of 3072. Note that we need to account for the teacher forward call that we do in our analysis.

**SALT**  The spatial resolution is $224 \times 224$ with a patch size of 16. We use a masking ratio of 0.9 for all stages in our training as we use the same multi-block masking for training teacher and student encoders. The rest of the details are identical to those described above for V-JEPA 2. The main difference between our method and V-JEPA 2 is the use of same-sized or smaller teacher encoder as well using a smaller resolution for the inputs. Together, these significantly lower the training compute requirements for SALT over V-JEPA 2.

Finally, we report the GPU-hours estimated by running training for a 20 steps on a single NVIDIA A100 GPU in Table 11. We observe a strong correlation between the ordering of models provided by GPU-hours versus that obtained from "6ND" FLOPs estimate.

Table 10: FLOPs estimate for VideoMAEv2, V-JEPA 2 and SALT models.

| Model | Input Resolution | $N_e$ (B) | $N_T$ (B) | $N_p$ (B) | $D_e$ ($\times 10^9$) | $D_p$ ($\times 10^9$) | $D_t$ ($\times 10^9$) | # Samples (B) | Total FLOPs ($\times 10^{21}$) |
|---|---|---|---|---|---|---|---|---|---|
| VideoMAEv2-g/14 | $16 \times 224 \times 224$ | 1.1 | — | 0.012 | 331.8 | 165.9 | — | 1.6 | 2.2 |
| V-JEPA 2 L/16 | $16 \times 256 \times 256$ | 0.303 | 0.303 | 0.022 | 302.0 | 3019.9 | 1510.2 | 0.7 | 1.9 |
| V-JEPA 2 H/16 | $16 \times 256 \times 256$ | 0.632 | 0.632 | 0.022 | 302.0 | 3019.9 | 1509.9 | 0.7 | 3.5 |
| V-JEPA 2 g/16 | $16 \times 256 \times 256$ | 1.012 | 1.012 | 0.022 | 302.0 | 3019.9 | 1509.9 | 0.7 | 5.3 |
| SALT-L/16 | $16 \times 224 \times 224$ | 0.3 | 0.303 | 0.022 | 154.1 | 1541.4 | 770.7 | 0.7 | 1.2 |
| SALT-H/16 | $16 \times 224 \times 224$ | 0.6 | 0.303 | 0.022 | 154.1 | 1541.4 | 770.7 | 0.7 | 1.5 |
| SALT-g/16 | $16 \times 224 \times 224$ | 1.0 | 0.303 | 0.022 | 154.1 | 1541.4 | 770.7 | 0.7 | 1.8 |
| SALT-G/16 | $16 \times 224 \times 224$ | 1.8 | 0.303 | 0.022 | 154.1 | 1541.4 | 770.7 | 0.7 | 2.6 |

Table 11: Training compute and GPU hours. We evaluate our models on **SSv2** with input of $16 \times 2 \times 3$ (*V-JEPA 2 uses a spatial resolution of $256 \times 256$, and SALT utilizes $224 \times 224$.). We compute TFLOPs under the same batch size and masking strategy, and measure on one single A100 GPU for all methods to ensure fairness and we exclude data-loading overhead and GPU-communication load from all measurements to ensure they are CPU-agnostic. The results in this table are used in Figure 1.

| Method | Teacher Params | Student Params | # Seen Samples ($\times 10^9$) | Total Compute ($\times 10^{21}$ FLOPs) | GPU-hrs | SSv2 Top-1 (%) | |
|---|---|---|---|---|---|---|---|
| | | | | | | $16 \times 2 \times 3$ | $16 \times 1 \times 1$ |
| V-JEPA 2 ViT-L Assran et al. (2025) | 302M | 302M | 7.4 | 1.9 | 9800 | 73.7 | 69.6 |
| V-JEPA 2 ViT-H Assran et al. (2025) | 631M | 631M | 7.4 | 3.5 | 14377 | 74.0 | 69.6 |
| V-JEPA 2 ViT-g Assran et al. (2025) | 1B | 1B | 7.4 | 5.3 | 18708 | 75.3 | 72.2 |
| SALT-Stage-1 ViT-L | N/A | 302M | 7.4 | 0.24 | 9062 | - | 66.2 |
| SALT ViT-L | **302M** | 302M | 7.4 | **1.2** | 8263 | 74.9 | 71.3 |
| SALT ViT-H | **302M** | 631M | 7.4 | **1.5** | 9574 | 75.4 | 72.6 |
| SALT ViT-g* | **302M** | 1B | 7.4 | **1.9** | 10476 | 76.2 | 72.9 |
| SALT ViT-G* | **302M** | 2B | 7.4 | **2.6** | 12379 | 76.1 | 73.2 |

# G    COMPUTE BUDGET SPECIFIED VIA FLOPS AND OPTIMIZATION STEPS

While we use FLOPs in our accuracy-compute trade-off analysis, we specify compute via the total number of optimization steps in our experiments. The use of the latter quantity is natural in our setup as the teacher's EMA update in V-JEPA 2 (Assran et al., 2025) depends on the student getting updated first which is similar to the nature of the update in our two-stage training as the teacher needs to be trained first followed by the student. The advantage with SALT is due from the fact that the teacher training is light-weight, and crucially, once a teacher model is trained, it may be used to train multiple student models.

## H  ADDITIONAL TABLES AND FIGURES

Table 12: V-JEPA 2 vs. SALT on same pretraining set. Kinetics-400 uses $16 \times 1 \times 1$ (number of frames in clip by temporal crops by spatial crops), Something-Something v2 (SSv2) uses $16 \times 1 \times 1$ while COIN is run with $16 \times 8 \times 3$. All models are evaluated using a spatial resolution of $224 \times 224$ pixels. The results in this table are used in Figure 2.

| Method | Teacher | Student | IN-1K | K400 | SSv2 | COIN | Diving-48 | Jester | Avg |
|---|---|---|---|---|---|---|---|---|---|
| V-JEPA 2 (w/ our dataset) | ViT-B | ViT-B | 66.9 | 66.9 | 61.4 | 68.4 | 71.0 | 95.9 | 71.8 |
| | ViT-L | ViT-L | 73.7 | 73.3 | 68.4 | 83.1 | 82.1 | 97.0 | 79.6 |
| | ViT-H | ViT-H | 76.7 | 73.6 | 68.9 | 84.9 | 84.5 | 97.1 | 81.0 |
| SALT Stage 1 (V-Pixel) | N/A | ViT-B | 70.3 | 65.2 | 60.9 | 73.3 | 72.9 | 95.7 | 73.1 |
| | N/A | ViT-L | 75.5 | 70.4 | 66.2 | 77.9 | 76.8 | 96.9 | 77.3 |
| SALT Stage 2 | ViT-B | ViT-B | 74.8 | 70.9 | 66.1 | 80.5 | 78.7 | 96.7 | 78.0 |
| | | ViT-L | 79.0 | 76.0 | 71.3 | 85.3 | 82.5 | 97.2 | 81.9 |
| | ViT-L | ViT-H | 79.6 | 77.2 | 72.6 | 87.0 | 86.4 | 97.3 | 83.4 |
| | | ViT-g | 79.7 | 78.0 | 72.9 | 87.0 | 85.5 | 97.4 | 83.4 |
| | | ViT-G | 80.3 | 78.9 | 73.2 | 87.5 | 85.3 | 97.4 | 83.8 |

Table 13: Ablation on teacher pretraining datasets. We test teacher and student model using frozen-backbone evaluation %). We report Top-1 accuracy on K400, SSv2, and ImageNet-1k, and COIN. This ablation is used in Figure 4.

| Dataset | # Samples | K400 | SSv2 | IN1k | COIN | Avg |
|---|---|---|---|---|---|---|
| | | **SALT-teacher** | | | | |
| **V-3.6M (default)** | 3,626,089 | 70.4 | 66.2 | 75.5 | 77.9 | 72.5 |
| K710 | 657,217 | 71.0 | 65.2 | 74.2 | 78.9 | 72.3 |
| Panda2.8M | 2,799,959 | 69.2 | 64.9 | 75.3 | 79.5 | 72.2 |
| SSv2 | 168,913 | 56.8 | 63.6 | 61.7 | 69.0 | 62.8 |
| K710 + SSv2 | 826,130 | 70.0 | 67.8 | 74.0 | 79.2 | 72.8 |
| ImageNet-1k | 1,281,167 | 51.9 | 39.3 | 80.6 | 67.4 | 59.8 |
| | | **SALT-student** | | | | |
| **V-3.6M (default)** | | 75.5 | 70.9 | 78.4 | 84.9 | 77.4 |
| K710 | | 75.7 | 70.6 | 78.4 | 85.0 | 77.4 |
| SSv2 | **V-3.6M (default)** | 72.9 | 69.8 | 76.2 | 83.1 | 75.5 |
| Panda2.8M | | 75.3 | 70.5 | 78.4 | 84.4 | 77.2 |
| K710 + SSv2 | | 75.1 | 71.1 | 78.0 | 84.9 | 77.3 |
| ImageNet-1k | | 72.1 | 66.5 | 79.1 | 82.0 | 74.9 |

## I  LLM USAGE STATEMENT

During manuscript preparation, LLMs were used to help with editing and polishing the draft.

Table 14: **Teacher model size ablation.** We report frozen-backbone Top-1 on K400, SSv2, IN1K, and COIN. The top block shows teachers evaluated directly. The lower blocks show students distilled from different teacher sizes (two student sizes: ViT-L and ViT-G). The data in this table is used in Figure 6.

| Model size | | K400 | SSv2 | IN1k | COIN | Avg |
|---|---|---|---|---|---|---|
| **Teacher** | **Student** | | | | | |
| ViT-B | — | 65.2 | 60.9 | 70.3 | 73.3 | 67.4 |
| **ViT-L (default)** | — | 70.4 | 66.2 | 75.5 | 77.9 | 72.5 |
| ViT-H | — | 71.1 | 67.4 | 76.7 | 81.5 | 74.2 |
| ViT-G | — | 73.6 | 68.5 | 77.4 | 81.4 | 75.2 |
| **SALT-student** | | | | | | |
| ViT-B | | 74.4 | 69.5 | 78.0 | 84.0 | 76.5 |
| **ViT-L (default)** | ViT-L | 75.5 | 70.9 | 78.4 | 84.9 | 77.4 |
| ViT-H | | 75.6 | 70.7 | 78.3 | 84.4 | 77.3 |
| ViT-G | | 75.5 | 71.5 | 78.5 | 84.7 | 77.6 |
| ViT-B | | 76.9 | 71.8 | 79.0 | 85.4 | 78.3 |
| **ViT-L (default)** | ViT-G | 77.6 | 71.9 | 79.3 | 87.0 | 79.0 |
| ViT-H | | 77.5 | 72.4 | 79.6 | 85.3 | 78.7 |

Table 15: **Masking strategy ablation.** We report frozen-backbone Top-1 on K400, SSv2, IN1k, and COIN; Top block: teachers evaluated directly. Bottom block: students (fixed student recipe) trained from different teachers. The data in this table is used in Figure 5.

| Teacher masking strategy | # masks | Masking ratio | Student Masking strategy | K400 | SSv2 | IN1k | COIN | Avg |
|---|---|---|---|---|---|---|---|---|
| SALT-teacher | | | | | | | | |
| **Long-short block mask (default)** | ×2 | ≈ 0.9 | — | 70.4 | 66.2 | 75.5 | 77.9 | 72.5 |
| Random tube | ×2 | [0.9, 0.9] | — | 69.2 | 64.7 | 74.1 | 74.9 | 70.7 |
| Random tube | ×1 | [0.9] | — | 67.9 | 63.3 | 72.6 | 73.5 | 69.3 |
| Causal mask | ×2 | [0.9, 0.9] | — | 47.4 | 34.3 | 57.3 | 58.8 | 49.5 |
| SALT-student | | | | | | | | |
| Long-short block mask (default) | ×2 | ≈ 0.9 | | 75.5 | 70.9 | 78.4 | 84.9 | 77.4 |
| Random tube | ×2 | [0.9, 0.9] | Long-short block mask | 75.0 | 70.3 | 78.1 | 84.3 | 76.9 |
| Random tube | ×1 | [0.9] | | 75.0 | 70.4 | 78.0 | 84.8 | 77.1 |
| Causal mask | ×2 | [0.9, 0.9] | | 71.7 | 66.3 | 75.0 | 81.7 | 73.7 |

Table 16: **Compute–accuracy tradeoffs at matched total steps.** Each block fixes the *total* pretraining steps (budget) and compares V-JEPA,2 to our two–stage schedule (teacher+student). FLOPs are reported as $\times 10^{21}$. Metrics are frozen–backbone Top-1 on K400, SSv2, IN1k, and COIN. This table includes data presented in Figure 7.

| Teacher Steps | Teacher FLOPs | Student Steps | Student FLOPs | Total FLOPs | Total Steps | K400 | SSv2 | IN1k | COIN | Avg |
|---|---|---|---|---|---|---|---|---|---|---|
| **Budget: 240k total steps** | | | | | | | | | | |
| **V-JEPA 2 (baseline)** | 1.900 | — | — | 1.900 | 240k | 73.3 | 68.4 | 73.7 | 83.1 | 74.6 |
| 80k+80k | 0.717 | 80k | 0.476 | 1.193 | 240k | 76.5 | 71.8 | 79.0 | 86.8 | 78.5 |
| 40k+80k | 0.597 | 120k | 0.714 | 1.311 | 240k | 76.8 | 71.7 | 79.3 | 86.7 | 78.6 |
| 80k | 0.241 | 160k | 0.951 | 1.192 | 240k | 76.0 | 71.3 | 79.0 | 85.3 | 77.9 |
| 40k | 0.121 | 200k | 1.190 | 1.311 | 240k | 76.3 | 71.7 | 79.1 | 85.6 | 78.2 |
| 20k | 0.061 | 220k | 1.309 | 1.370 | 240k | 75.8 | 71.2 | 78.7 | 84.9 | 77.7 |
| **Budget: 160k total steps** | | | | | | | | | | |
| **V-JEPA 2 (baseline)** | 1.260 | — | — | 1.260 | 160k | 73.5 | 68.3 | 74.8 | 82.9 | 74.9 |
| 80k | 0.241 | 80k | 0.476 | 0.717 | 160k | 75.5 | 70.9 | 78.4 | 84.9 | 77.4 |
| 40k | 0.121 | 120k | 0.714 | 0.835 | 160k | 75.5 | 70.9 | 78.4 | 85.4 | 77.6 |
| 20k | 0.061 | 140k | 0.833 | 0.894 | 160k | 74.9 | 70.6 | 78.0 | 84.6 | 77.0 |
| **Budget: 120k total steps** | | | | | | | | | | |
| **V-JEPA 2 (baseline)** | 0.951 | — | — | 0.951 | 120k | 73.3 | 67.9 | 74.7 | 81.5 | 74.4 |
| 80k | 0.241 | 40k | 0.238 | 0.479 | 120k | 74.0 | 69.5 | 77.7 | 84.5 | 76.4 |
| 40k | 0.121 | 80k | 0.476 | 0.597 | 120k | 75.0 | 70.2 | 78.0 | 84.9 | 77.0 |
| 20k | 0.061 | 100k | 0.595 | 0.656 | 120k | 74.6 | 70.1 | 77.5 | 84.5 | 76.7 |

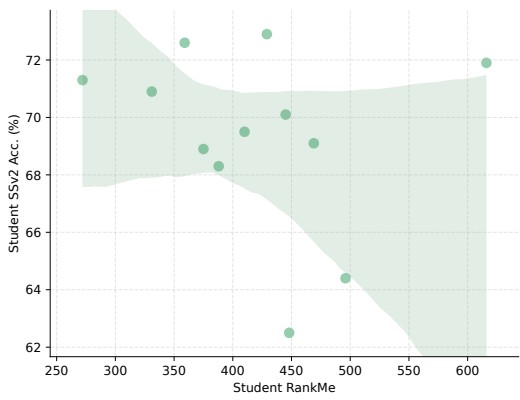

Figure 10: Correlation between RankME and downstream accuracy. We use the same SALT-Stage-1-80k teacher checkpoint.

