# OpenReview forum: "Rethinking JEPA: Compute‑Efficient Video Self-Supervised Learning with Frozen Teachers"
_ICLR.cc/2026/Conference — ICLR 2026 Poster_

### Official Review · Reviewer_k4fL · 2025-10-28

**Soundness:** 3
**Presentation:** 4
**Contribution:** 3
**Rating:** 6
**Confidence:** 4

**Summary:**

The paper proposes a modification to the V-JEPA framework, which consists of two stages. In the first stage, a teacher network is pretrained on a dataset—VMAE-v2 in this case. In the second stage, a significantly larger student network is trained to predict the teacher’s features using the V-JEPA loss function. Since the teacher is considerably smaller than the student, the total computational cost is dominated by the student. Nevertheless, this approach achieves higher performance and converges faster than the original V-JEPA framework, which trains the teacher and student jointly via EMA, under the same computational budget.

**Strengths:**

1. The idea of employing a smaller teacher than the student is both interesting and computationally efficient.
2. The method’s two-stage design simplifies training and is likely to offer greater stability compared to EMA-based teacher frameworks such as V-JEPA.
3. The approach is supported by an extensive set of experiments ensuring fair comparisons in terms of total compute and performance.
4. The method is evaluated across diverse settings, including image and video datasets as well as intuitive physics understanding, providing a comprehensive assessment of its effectiveness.

**Weaknesses:**

1. The main motivation behind the proposed framework is to address the instability of V-JEPA, which reportedly requires meticulous hyperparameter tuning. However, the paper does not provide empirical evidence to support this claim. Is your model more stable with respect to the training hyperparameters listed in Table 5? It would strengthen the argument to include a comparison between V-JEPA and SALT in terms of stability across those hyperparameters.

2. The paper does not include an ablation study on the impact of teacher pretraining. How sensitive is the proposed method to the choice of the pretrained teacher? What is the rationale for using VMAE-v2 specifically? I encourage the authors to include an ablation for this factor, at least for a ViT-B teacher. For example, a teacher could be selected from one of the following works: [1, 2, 3, 4].

3. The concept of training from a frozen teacher network has been previously explored in [2, 4], although in those works the teacher is an image model. It would be fair to discuss these approaches in the related work section, as the underlying idea is conceptually related to the current paper.

4. Is there a particular reason why the proposed method is only evaluated on video datasets? Was it also tested on images, and if so, did it perform comparably? Including such an analysis could clarify whether the idea of using a smaller teacher than the student is specific to video settings.

---

**References**

1. *Masked Motion Encoding for Self-Supervised Video Representation Learning*, CVPR 2023
2. *SIGMA: Sinkhorn-Guided Masked Video Modeling*, ECCV 2024
3. *MGMAE: Motion Guided Masking for Video Masked Autoencoding*, ICCV 2023
4. *SMILE: Infusing Spatial and Motion Semantics in Masked Video Learning*, CVPR 2025

**Questions:**

Please refer to weaknesses. I tried to provide clear questions there.

---

> ### Author Response · Authors · 2025-11-20
> **Reply to Reviewer k4fL**
>
> ---
> ## Q1:  Instability of V-JEPA and empirical evidence.
>
> ## A1:
> Thanks for raising this point. Our motivation follows the broader SSL literature where EMA-based methods (e.g., BYOL/DINO/V-JEPA) are known to require careful tuning of momentum, temperature, and other hyperparameters to avoid collapse. SALT's main contribution is showing that a simplified recipe actually works better by removing the necessity of EMA and stopgrad: both stages just minimize simple losses (pixel reconstruction and masked latent prediction), and we show that the student loss correlates well with downstream performance, which makes training and model selection easier in practice. We do not provide a full hyperparameter sweep (Table 5) due to computational cost. In the revision, we have removed the strong wording about V-JEPA being unstable in Section 2.2.
>
> ---
>
> ## Q2: Which teacher did you use and how it was pretrained, and why did you choose VMAE-v2 instead of other possible teachers (e.g., [1–4])? Can you show an ablation (e.g., with different ViT-B teachers) to prove SALT is not overly sensitive to this choice?
>
> ## A2:
> Thanks for the detailed comments. For all main results (Table 1) and ablations (Fig. 4–6), we train the teacher **from scratch** for 80k steps on the same V-3.6M dataset used for the students. We do **not** use a pretrained VMAE-v2 or any other off-the-shelf model as the teacher. This is intentional: we want to study teacher **properties in a fair and controllable setting**, where data, recipe, and compute are matched, rather than relying on stronger pretrained models whose own training cost and setup are not directly comparable.
>
> Under this setup, we carefully ablate (i) teacher pretraining data (Fig. 4), (ii) teacher size (ViT-B/L/H/G; see Fig. 6 and App. Table 12 for the ViT-B teacher), and (iii) teacher training schedule. These experiments show that SALT is robust to the teacher’s pre-training. While using off-the-shelf teachers from [1–4] is an interesting extension, they are not compute-free and are not good candidates for fair comparison, so we leave that to future work.
>
> ---
>
> ## Q3: [2, 4] related work
>
> ## A3:
> Thanks for bringing up these works. We have added the discussion in the related works.
>
> ---
>
> ## Q4: Only evaluated on video datasets? Was it also tested on images?
> ## A4:
> Our main focus in this work is video representation learning. To check whether the learned representations also transfer to appearance/scene recognition, we already include frozen evaluation on ImageNet-1K: in Table 13, SALT’s ViT-L student reaches 79.0% vs 73.7% for V-JEPA 2 under the same protocol, indicating that our design is not specific to video-only downstream tasks. We did not train an image-only version of SALT in this paper, and we will clarify this scope and mention a dedicated image-only study as interesting future work.
>
> ---
>
> **References**
>
> [1] Sun, et al. "Masked Motion Encoding for Self-Supervised Video Representation Learning." CVPR 2023.
> [2] Salehi, et al. "SIGMA: Sinkhorn-Guided Masked Video Modeling." ECCV 2024.
> [3] Huang, et al. "MGMAE: Motion Guided Masking for Video Masked Autoencoding." ICCV 2023.
> [4] Thoker, et al. "SMILE: Infusing Spatial and Motion Semantics in Masked Video Learning." CVPR 2025.

---

### Official Review · Reviewer_R6kh · 2025-10-30

**Soundness:** 3
**Presentation:** 3
**Contribution:** 2
**Rating:** 4
**Confidence:** 4

**Summary:**

The paper revisits video self-supervised learning under the JEPA framework and introduces SALT , which replaces the EMA-updated teacher with a frozen one.
By decoupling teacher and student training into pixel reconstruction and masked latent prediction, the method improves efficiency, transparency, and scalability. Experiments show that SALT outperforms V-JEPA2 at matched compute while remaining robust to teacher quality.

**Strengths:**

1.The method consists of two stages: pixel reconstruction and masked latent prediction, allowing small teacher models to train large-scale student models. Each training stage has a clear loss function with physical meaning, enabling SALT to avoid representation collapse.
2.On multiple mainstream video understanding benchmarks, SALT consistently outperforms V-JEPA2 with the same model architecture, particularly excelling on motion understanding tasks (SSv2), demonstrating stronger temporal modeling capabilities.
3.SALT can successfully train larger student models using smaller teacher models, improving computational efficiency and scalability.

**Weaknesses:**

1.Although the teacher is frozen, a pixel reconstruction training stage is still required first, introducing an additional pretraining step and making the process slightly longer compared to fully end-to-end methods.
2.While SALT performs exceptionally on temporal tasks (SSv2), its improvements on appearance-based tasks (such as K400) are modest, and it still lags slightly behind the latest methods (PEcoreG, InternVideo2-1B).
3.Although the overall computational cost is lower, SALT’s two-stage process still requires training the teacher model first, increasing the overall implementation complexity and making the workflow slightly longer compared to end-to-end EMA-based methods.
4.Missing code reduces reproducibility.

**Questions:**

1.Although SALT claims lower computational cost than the EMA baseline, it still requires training the teacher before the student. Compared to end-to-end EMA methods, does it truly save overall training time?
2.Experiments show that SALT achieves significant improvements on SSv2, but only limited gains on K400. Is SALT more suited for temporal modeling? If the task primarily involves appearance or scene recognition, does the method still maintain a clear advantage?
3.SALT relies on a two-stage teacher-student training, which adds some training complexity. Additionally, its performance gains on the K400 dataset are modest and fall short compared to the latest methods (PEcoreG, InternVideo2-1B).

---

> ### Author Response · Authors · 2025-11-20
> **Reply to Reviewer R6kh**
>
> ---
> ## Q1: Two-stage training adds extra steps and concerns about training time.
>
> ## A1:
> Thanks for raising this concern. We would like to clarify how we allocate compute and training time in our comparisons. Following V-JEPA 2, we **fix the total number of training steps and the batch size** to be the same across methods. For example, in Table 1, both V-JEPA 2 and SALT use 240k steps with the same batch size; SALT simply splits this into **80k steps for the teacher + 160k steps for the student**.
>
> Regarding training time, our method is indeed faster than EMA-based training.
> Although EMA updates do not require gradients, the EMA teacher still processes the full video and, by design, must have the same size as the student. In SALT, the teacher is frozen and decoupled from the student, so we can use a much smaller teacher, which substantially reduces forward computation (see Appendix F and Table 10).
> Moreover, our Stage-1 training does not involve EMA updates and is naturally faster. Taken together, these factors allow SALT to achieve better overall efficiency, with the savings becoming even more evident as we scale up the student model size. Finally, the wall-clock GPU times reported in Table 11 are consistent with this analysis.
>
> ---
>
>
> ## Q2: Experiments show that SALT achieves significant improvements on SSv2, but only limited gains on K400. Is SALT more suited for temporal modeling? If the task primarily involves appearance or scene recognition, does the method still maintain a clear advantage?
>
>
> ## A2:
> Thanks for raising this concern. As the reviewer correctly points out, SALT lags behind methods like PEcoreG and InternVideo2-1B while improving over V-JEPA 2, albeit by a small amount. The latter point supports our main thesis that our design decision to remove EMA is effective at learning representations that are useful for appearance and temporal understanding while improving computational efficiency.
>
> Furthermore, SALT also brings clear gains on appearance-centric tasks when compared under the **same pre-training data and compute**. In Fig. 2(a), when V-JEPA 2 and SALT are trained on the same dataset with the same number of steps, SALT reaches **76.0%** top-1 on K400 vs **73.3%** for V-JEPA 2. Similarly, on the pure scene-recognition task ImageNet-1K, SALT improves over V-JEPA 2 by **+5.3%**. These results indicate that SALT is not only effective on temporal-centric benchmarks (SSv2) but also yields substantial improvements on appearance-centric ones when compared to our primary baseline.
>
> Lastly, regarding PEcoreG and InternVideo2-1B, we acknowledge that SALT is slightly behind these latest models on K400. Our focus is purely self-supervised, video-only pretraining under controlled compute and data. Given that SALT achieves competitive performance while significantly simplifying training and improving efficiency, we view SALT as broadly applicable, not only to temporal modeling but also to appearance- and scene-centric tasks.
>
>
> ---
>
>
> ## Q3: Overall implementation complexity.
>
>
> ## A3:
> Thanks for raising this issue. Although SALT uses a two-stage schedule, it does **not** increase implementation complexity. In practice, SALT actually **simplifies** training compared to V-JEPA 2: we remove the EMA update mechanism, stop-gradient, and teacher momentum synchronization. Both stages can be implemented within essentially the same training loop by switching the loss (pixel reconstruction in Stage 1, masked latent prediction in Stage 2) and freezing the teacher. In terms of code, the number of lines and the overall pipeline are very similar, but without the extra EMA update.
>
>
> ---
>
>
> ## Q4: Reproducibility.
>
>
> ## A4:
> We are looking into releasing code for reproducibility.

---

### Official Review · Reviewer_ZRSW · 2025-10-31

**Soundness:** 3
**Presentation:** 3
**Contribution:** 3
**Rating:** 6
**Confidence:** 4

**Summary:**

This paper introduces SALT (Static-teacher Asymmetric Latent Training), a two-stage, unregularized framework for video representation pretraining. The authors challenge the conventional approach of using dynamic teachers in self-distillation architectures, such as V-JEPA, and propose that a less optimal, pre-trained teacher is sufficient for distilling a student model. Specifically, they pretrain a teacher model using a pixel reconstruction objective, freeze the teacher model, and then use it to distill the student, demonstrating performance improvements over V-JEPA under fair settings. Extensive experiments validate the effectiveness of SALT and provide justification for its design choices.

**Strengths:**

1. The paper presents an alternative to traditional V-JEPA, which relies on EMA and stop-gradient operations.
2. The proposed SALT method demonstrates superior performance and greater efficiency compared to V-JEPA.
3. The experiments are thorough and well-executed, including ablation studies that effectively validate the contributions of each design element in the proposed method.

**Weaknesses:**

1. The VJEPA framework utilizes EMA-based self-distillation, where performance improvement is driven by the joint optimization of both the teacher and student models. However, it seems counterintuitive that SALT employs a smaller-scale or less capable teacher to distill a more performant student. This discrepancy warrants a more detailed analysis and clarification.
2. The authors claim that "a high-performing teacher is not necessary to train a high-quality student." This assertion appears to hold under the assumption of fixed total training steps across two stages. However, in an unconstrained setting, does the performance of the teacher model positively correlate with that of the student? Specifically, if teachers are trained for 20k, 40k, and 80k steps, respectively, and students are trained for 80k steps in all cases, would the performance of the models improve sequentially?
3. The proposed method primarily relies on pretraining and distillation, while it seems that not explicitly present temporal optimization. Recent advancements in image-to-video distillation [1-3] have incorporated temporal relationships into strong representation models during distillation. The authors are encouraged to compare their approach with these recent methods in the related work section, in order to investigate how temporal information can be leveraged in representation distillation.

[1] Liu Y, Xu Q, Wen P, et al. When the future becomes the past: Taming temporal correspondence for self-supervised video representation learning. In CVPR, 2025.
[2] Li R, Liu D. Spatial-then-temporal self-supervised learning for video correspondence. In CVPR, 2023.
[3] Hu Y, Wang R, Zhang K, et al. Semantic-aware fine-grained correspondence. In ECCV, 2022.

**Questions:**

Please refer to the weaknesses.

---

> ### Author Response · Authors · 2025-11-20
> **Reply to Reviewer ZRSW**
>
> ---
>
> ## Q1: Why can a smaller / less capable teacher train a stronger student?
>
> ## A1:
>   Thanks for raising this point. First, conceptually, SALT follows the masked-prediction distillation paradigm rather than classical logit distillation. In our framework, the student is **not** trained to match the teacher’s logits or minimize the teacher’s loss; instead, it learns to **predict the teacher’s masked embeddings**.  Second, empirically, our controlled experiments (Sec. 5.3, App. E.3) show that (i) increasing the teacher from ViT-L to larger variants yields only marginal or no gains for the student, and (ii) the teacher model’s metrics such as pretraining loss and RankMe [4] (embedding quality) do not reliably predict student frozen accuracy (Fig. 9). These results indicate that after the teacher reaches a **moderate quality threshold**, its role is to simply provide a stable, “good-enough” target.  Thus, increasing the teacher size is not **efficient** for improving the student’s downstream performance, and the student’s final performance is largely determined by its own size and training budget. Our claim that “a high-performing teacher is not necessary” is thus made explicitly in this **compute-aware** regime, where smaller or partially trained teachers offer a better accuracy/compute trade-off than very large teachers.
>
>
> ---
>
>
> ## Q2: On the claim “a high-performing teacher is not necessary” in an unconstrained setting.
>
> ## A2:
> Thanks for this clarification question. Our claim that “a high-performing teacher is not necessary to train a high-quality student” is made in the **compute-aware, two-stage setting** we study, where the total training budget is fixed and must be split between teacher and student. In this regime, our experiments in Sec. 5.3 and App. E.3 shows that once the teacher reaches a moderate quality threshold, allocating more compute to the teacher yields very limited student gains compared to spending that compute on the student.
>
> Following the reviewer’s suggestion, we additionally ran an experiment in an *unconstrained* setting: we train ViT-L teachers for **20k / 40k / 80k** steps, and then train the same size students for **80k** steps in all three cases. We also list the fixed teacher’s compute but train the student for **20k / 40k / 80k**  comparison. The results can be found in the tables below. First, we observe that increasing either teacher or student training steps improves performance. Second, the effect of allocating more compute to the student is much stronger: for the same increase from 20k to 80k steps, scaling the teacher yields only a small gain **+1.1%** in average accuracy), whereas scaling the student yields a much larger improvement (**+5%**). This supports our choice of using a relatively small, moderately trained teacher and allocating most of the compute to the student for better efficiency.
>
>
> | Teacher | Student | Total steps | K400 | SSV2 | In1k | COIN | Avg  |
> |--------:|--------:|------------:|-----:|-----:|-----:|---------:|-----:|
> | 20K     | 80K     | 100K        | 73.1 | 69.2 | 76.2 | 83.5     | 75.5 |
> | 40K     | 80K     | 120K        | 74.3 | 69.5 | 76.9 | 83.7     | 76.1 |
> | 80K     | 80K     | 160K        | 74.9 | 70.2 | 77.5 | 83.9     | 76.6 |
>
>
> | Teacher | Student | Total steps | K400 | SSV2 | In1k | COIN | Avg  |
> |--------:|--------:|------------:|-----:|-----:|-----:|---------:|-----:|
> | 80K     | 20K     | 100K        | 69.9 | 65.6 | 73.2 | 77.8     | 71.6 |
> | 80K     | 40K     | 120K        | 73.2 | 68.8 | 76.5 | 82.5     | 75.3 |
> | 80K     | 80K     | 160K        | 74.9 | 70.2 | 77.5 | 83.9     | 76.6 |
>
>
>
> ---
>
>
> ## Q3: Related works.
>
> ## A3:
> We thank the reviewer for pointing out these relevant works. We have added these under related works in our updated version.

---

### Official Review · Reviewer_3hkH · 2025-11-01

**Soundness:** 3
**Presentation:** 3
**Contribution:** 2
**Rating:** 6
**Confidence:** 5

**Summary:**

This paper introduces SALT (Static-teacher Asymmetric Latent Training), a two-stage framework for compute-efficient self-supervised video representation learning. The method challenges the necessity of the complex, online, EMA-updated teacher used in state-of-the-art models like Video Joint Embedding Predictive Architectures (V-JEPA). The core design of this work is borrowed from Masked Video Distillation, using two-stage training: (1) At stage-1, a teacher encoder is trained using a simple pixel-reconstruction objective, similar to VideoMAE; (2) At stage-2, the teacher encoder is frozen, and a student encoder, along with a predictor, is trained to predict the teacher's latent representations for masked regions of the input video. The results of experiments show that high-performing student models can be trained using smaller teachers—models. The main results show that SALT outperforms the strong V-JEPA 2 baseline on multiple video understanding benchmarks (e.g., Something-Something-v2, Kinetics-400), especially when matched for computational cost.

**Strengths:**

1. The experimental validation is extensive and rigorous. The systematic comparison against strong baselines, including a retrained V-JEPA 2, convincingly demonstrates SALT's superiority in both accuracy and efficiency. The paper includes a comprehensive set of ablations that analyze the impact of key design choices, such as the teacher's training data, masking strategy, model size, and the compute allocation between teacher and student.

2. The demonstration that the student's training loss is highly predictive of downstream performance (Figure 3) is a major practical advantage.

3. This paper is very well-written, clearly structured, and easy to follow.

**Weaknesses:**

1. **Limited Novelty in the Self-Supervised Learning Framework**: The proposed two-stage distillation framework, which uses a frozen teacher to provide target features, shares a significant conceptual overlap with prior work. Specifically, the core idea was proposed and studied in Masked Video Distillation (MVD)[1] two years ago and has been scaled and improved in subsequent works like VideoPrism[2] and InternVideo2[3]. Therefore, the method's design is not an entirely novel concept. The paper's primary contributions are better framed as a large-scale empirical study that (a) provides a fair and direct comparison against a strong momentum-encoder baseline (V-JEPA 2) on larger datasets and models, and (b) uncovers some valuable insights like smaller teachers can produce strong students. Key conclusions related to the method's design (such as the student outperforming the teacher or a frozen teacher being superior to an EMA teacher) are consistent with findings in these earlier studies. It would be beneficial for the authors to more precisely position their contributions in the introduction, emphasizing the scientific insights and empirical discoveries rather than the originality of the algorithmic framework itself.
2. **Lack of Deeper Analysis on Teacher Model Size Saturation**: The experiments in Figure 6 suggest that, under the paper's settings, increasing the teacher's model size does not yield significant performance gains for the student. However, the paper does not offer a deep theoretical or experimental analysis to explain this phenomenon. For instance, an analysis of the target features provided by teachers of different sizes, or an investigation into how this finding correlates with the student's model size, would provide much greater insight into the dynamics of the teacher-student relationship in this framework.
3. **Unexplored Performance Bottleneck at Larger Scales**: Table 1 reveals a slight performance degradation on the SSv2 benchmark as the SALT model is scaled from 1B to 2B parameters. The paper does not address or analyze this observation. This trend suggests that SALT may face a performance bottleneck at very large scales, and an investigation into the potential causes would significantly strengthen the paper's contribution to understanding the scalability of this approach.

[1] Wang, et al. "Masked video distillation: Rethinking masked feature modeling for self-supervised video representation learning." CVPR 2023.

[2] Zhao, Long, et al. "Videoprism: A foundational visual encoder for video understanding." arXiv preprint arXiv:2402.13217 (2024).

[3] Wang, Yi, et al. "Internvideo2: Scaling foundation models for multimodal video understanding." ECCV 2024.

**Questions:**

1. The V-JEPA paper reported frozen evaluation results using a pretrained model from MVD. Given that this represents a highly relevant baseline for a frozen-teacher approach, the authors should include the results of this baseline in the main results table.

2. Have the authors considered the possibility of iterating the training process? Specifically, could the student model trained in Stage 2 be frozen and used as a new teacher to train a subsequent, potentially stronger, student model? This appears to be a natural extension of the method.

---

> ### Author Response · Authors · 2025-11-20
> **Reply to Reviewer 3hkH**
>
> ---
>
> ## Q1: Limited Novelty in the Self-Supervised Learning Framework.
>
>
> ## A1:
> We thank the reviewer for recognizing our findings and for the helpful suggestions. We agree that our use of a frozen teacher is conceptually related to [1, 2, 3], but we would like to clarify the differences and our specific contributions. First, these prior works either overlook **frozen-encoder evaluation** (e.g., MVD relies on full fine-tuning) or rely on an **off-the-shelf multimodal, contrastive teacher** such as CLIP [2, 3]; whereas our goal is to learn representations without recourse to external foundation models. In contrast, our work instantiates a **video-only JEPA-style framework** with a frozen teacher, demonstrating that EMA and stop-grad machinery can be removed.
>
> Second, we provide a **large-scale, compute-aware empirical study** that systematically quantifies how teacher size/quality and teacher–student compute allocation affect student performance in this setting. This directly addresses the open question of what constitutes a “good enough” teacher for self-supervised video learning.
>
> Finally, our contribution is not limited to this empirical insight: we also provide a **simple and efficient training recipe** that allows the community to train a state-of-the-art video SSL encoder using only straightforward loss optimization, without EMA updates or extra regularization tricks. In the revised introduction, we have updated L59–L84 to explicitly frame the paper around these scientific and practical insights.
>
>
> ---
>
>
> ## Q2: Lack of Deeper Analysis on Teacher Model Size Saturation.
>
> ## A2:
> Thanks for pointing this out. We indeed analyze this phenomenon empirically: our controlled experiments (Sec. 5.3, App. E.3) show that (i) increasing the teacher from ViT-L to larger variants yields only marginal or no gains for the student, and (ii) the teacher model’s metrics such as pretraining loss and RankMe [4] (embedding quality) do not reliably predict student frozen accuracy (Fig. 9). These results indicate that after the teacher reaches a **moderate quality threshold**, its role is to simply provide a stable, “good-enough” target.  Thus, increasing the teacher size is not **efficient** for improving the student’s downstream performance, and the student’s final performance is largely determined by its own size and training budget. Our claim that “a high-performing teacher is not necessary” is thus made explicitly in this **compute-aware** regime, where smaller teachers offer a better accuracy/compute trade-off than larger teachers. We believe theoretical analysis is an important direction for future work that would further strengthen this conclusion.
>
> ---
>
>
> ## Q3: Unexplored Performance Bottleneck at Larger Scales.
>
> ## A3:
> Thanks for the comment. First, we emphasize that our focus is to demonstrate that a ViT-L teacher (**up to 6 $\times$ smaller**) can still yield a state-of-the-art ViT-G student, which is **2 $\times$ larger** than the V-JEPA 2 ViT-g while also being roughly **2 $\times$ faster** to train. This strongly supports our claim that **a small teacher is sufficient** in this setting. Regarding the observed saturation, we hypothesize that it is driven primarily by data scale, rather than an inherent bottleneck of SALT. In all our scaling experiments, we increase the student model size while keeping the pretraining dataset fixed to V-3.6M. A study that jointly scales the model and data is an important direction for understanding SALT’s scalability. We have added this limitation in our latest manuscript in Section 7.
>
> ---
>
>
> ## Q4: Include MVD [1] results.
>
> ## A4:
> Thanks for pointing that out. We have included MVD results from Table 6 in V-JEPA  in our main manuscript.
>
> ---
>
>
> ## Q5: Iterative teacher.
>
> ## A5:
> Thanks for the constructive suggestion. Following it, we ran an additional experiment where stage 1 uses an 80k-step V-Pixel ViT-L model as the teacher to train a student for 80k steps. We then freeze this student and reuse it as the new teacher to train a final model for 80k steps from scratch. The final student‘s downstream results are reported in the table below.  We find that this **iterative SALT** variant yields a small but consistent improvement from 81.9 to 82.5 (+0.6) on average. Given the modest gains, we regard these results as preliminary and therefore keep a two-stage design in this work, leaving a systematic study of iterative distillation to future work.
>
> | Student | Teacher  | # Stages | K400 | SSV2 | Diving | Jester | In1k | COIN | Avg  |
> |---------|---------:|---------:|-----:|-----:|-------:|-------:|-----:|-----:|-----:|
> | 160k    | 80k      | **2**    | 76.0 | 71.3 | 82.5   | 97.2   | 79.0 | 85.3 | 81.9 |
> | 80k     | 80k+80k  | **3**    | 76.5 | 71.8 | 83.4   | 97.3   | 79.0 | 86.8 | 82.5 |

---

> > ### Author Response · Authors · 2025-11-20
> > **References**
> >
> > –-
> >
> > **References**
> >
> > [1] Wang, et al. "Masked video distillation: Rethinking masked feature modeling for self-supervised video representation learning." CVPR 2023.
> >
> > [2] Zhao, Long, et al. "Videoprism: A foundational visual encoder for video understanding." arXiv preprint arXiv:2402.13217 (2024).
> >
> > [3] Wang, Yi, et al. "Internvideo2: Scaling foundation models for multimodal video understanding." ECCV 2024.
> >
> > [4] Garrido, Quentin, et al. "Rankme: Assessing the downstream performance of pretrained self-supervised representations by their rank." ICML 2023.

---

### Author Response · Authors · 2025-11-27
**Authors-Reviewers Discussion**

Dear Reviewers and AC,

We thank you for the time you have spent reviewing our paper and providing feedback and asking questions. We have taken the reviews very seriously and conducted additional work to respond to questions raised in the initial reviews. We hope that our responses either address the issues raised by the reviewers or help the reviewers engage with us so that we can provide answers or clarifications as needed. We look forward to engaging with the reviewers during the remainder of the Authors-Reviewers Discussion Period.

Best,

Authors 15202

---

### Comment · Area_Chair_ujPY · 2025-11-27

Dear Reviewers,

As we enter the discussion phase, I strongly encourage you to read the authors' rebuttal carefully and acknowledge their effort. Silence is the worst outcome for an author. Even if the rebuttal does not change your final rating, a brief response explaining why the concerns remain unaddressed is crucial for a fair process. Please help us make an informed decision by engaging in a constructive dialogue.

AC

---

### Public Comment · ~Egor_Salygin1 · 2026-03-17

Thank you for the paper, this is a very interesting and timely contribution. I’m wondering whether the authors plan to release code, training configurations, and/or pretrained checkpoints for SALT. Having public artifacts would be extremely helpful for reproducibility and for follow-up research by the community. If a release is planned, could you share where it will be hosted and, if possible, an approximate timeline?

---

### Meta-Review · Area_Chair_Xyso · 2026-01-07

**Summary:**

Reviewers praised rigorous large-scale experiments, fair V-JEPA 2 comparisons, strong compute-efficiency/scalability, clear writing, and stable two-stage design. Concerns on novelty (overlap with frozen-teacher works like MVD/VideoPrism/InternVideo2), teacher-size saturation analysis, large-scale bottlenecks, and minor gaps were largely addressed in rebuttal via clarifications, new experiments (unconstrained-compute, iterative-teacher, MVD baseline), expanded discussions, and ablations. The strengths outweigh remaining concerns (need for deeper theory on weak teacher-student correlation; scalability limits at extreme scales), and reviewer consensus was reached during the discussion period.

(Citation optional) To further strengthen related work and contextualize staged/asymmetric training, authors may refer prior works like SimSiam (arXiv:2011.10566, staged training experiments) and AsymSiam (arXiv:2204.00613, asymmetric teacher-student relations study) for insights.

**Reviewer Concerns:**

Addressed:

- Novelty/positioning: Authors clarified large-scale, compute-aware video-only focus and differences from prior work.
- Teacher-size saturation and deeper dynamics: Added evidence, unconstrained-compute experiment, and iterative-teacher results.
- Large-scale degradation and bottlenecks: Explained trends and added MVD baseline.
- Motivation, ablations, related work, temporal aspects: Expanded discussion, added ablations, aligned motivation, highlighted IN-1K transfer.
- Perceived complexity/training time: Clarified matched compute and faster wall-clock time.

Still outstanding:

- Deeper theoretical/experimental explanation of why teacher quality weakly correlates with student performance beyond moderate thresholds.
- More thorough analysis of potential scalability limits at very large models.

**Reviewer Scores:**

- Reviewer 3hkH: Would likely remain 6 (concerns largely addressed, stable post-rebuttal).
- Reviewer ZRSW: Would likely remain 6 (positive, concerns addressed).
- Reviewer R6kh: Already increased to 6; would stay 6.
- Reviewer k4fL: Would likely remain 6 (positive, concerns addressed).

---

### Decision · Program_Chairs · 2026-01-26

Accept (Poster)